# Revisions of the Phenomenological and Statistical Statements of the Second Law of Thermodynamics

**DOI:** 10.3390/e26121122

**Published:** 2024-12-22

**Authors:** Grzegorz Marcin Koczan, Roberto Zivieri

**Affiliations:** 1Department of Mechanical Processing of Wood, Warsaw University of Life Sciences, 02-787 Warsaw, Poland; 2Istituto Nazionale di Alta Matematica (INdAM), 00185 Rome, Italy; roberto.zivieri@unife.it

**Keywords:** *perpetuum mobile* of the third kind, Carnot engine, heat–temperature inequality, ideal-gas entropy, reversibility paradox, Boltzmann H theorem, fluctuation theorems, probabilistic scheme of second law of thermodynamics

## Abstract

The status of the Second Law of Thermodynamics, even in the 21st century, is not as certain as when Arthur Eddington wrote about it a hundred years ago. It is not only about the truth of this law, but rather about its strict and exhaustive formulation. In the previous article, it was shown that two of the three most famous thermodynamic formulations of the Second Law of Thermodynamics are non-exhaustive. However, the status of the statistical approach, contrary to common and unfounded opinions, is even more difficult. It is known that Boltzmann did not manage to completely and correctly derive the Second Law of Thermodynamics from statistical mechanics, even though he probably did everything he could in this regard. In particular, he introduced molecular chaos into the extension of the Liouville equation, obtaining the Boltzmann equation. By using the *H* theorem, Boltzmann transferred the Second Law of Thermodynamics thesis to the molecular chaos hypothesis, which is not considered to be fully true. Therefore, the authors present a detailed and critical review of the issue of the Second Law of Thermodynamics and entropy from the perspective of phenomenological thermodynamics and statistical mechanics, as well as kinetic theory. On this basis, Propositions 1–3 for the statements of the Second Law of Thermodynamics are formulated in the original part of the article. Proposition 1 is based on resolving the misunderstanding of the *Perpetuum Mobile* of the Second Kind by introducing the *Perpetuum Mobile* of the Third Kind. Proposition 2 specifies the structure of allowed thermodynamic processes by using the Inequality of Heat and Temperature Proportions inspired by Eudoxus of Cnidus’s inequalities defining real numbers. Proposition 3 is a Probabilistic Scheme of the Second Law of Thermodynamics that, like a game, shows the statistical tendency for entropy to increase, even though the possibility of it decreasing cannot be completely ruled out. Proposition 3 is, in some sense, free from Loschmidt’s irreversibility paradox.

## 1. Introduction

### 1.1. General Review


*Today we are faced with the task of making the limits of validity of the Second Law of Thermodynamics more precise.*
Marian Smoluchowski (1914) [1]

The Second Law of Thermodynamics is one of the most remarkable physical laws for its profound meaning, its deep implications in every field of physics and the several applications in other fields of science, such as any field of engineering, chemistry, biology, genetics, medicine and, more generally, in natural sciences. And yet, behind Smoluchowski above motto there are grounds for examining the limits of this law’s validity [1].

In simple terms, Clausius’ First Statement of the Second Law of Thermodynamics states that the transfer of heat from a higher-temperature reservoir to a lower-temperature reservoir is spontaneous and continues until thermal equilibrium is established (if possible). Instead, the transfer of heat from a lower-temperature reservoir to a higher-temperature reservoir is not spontaneous and occurs only if work is performed [2]. More profoundly, the understanding of heat, energy and their interplay requires the knowledge of both the First Law of Thermodynamics and the Second Law of Thermodynamics [3].

Thanks to the introduction of the concept of entropy (*S*) by Clausius, who studied the relation between heat transfer and work as a form of heat energy, a second version of the Second Law of Thermodynamics can be stated, which can be identified with Clausius’ Second Statement for closed and isolated thermodynamic systems subjected to reversible processes in terms of entropy change: entropy does not decrease, namely, ΔS≥0. The entropy variation ΔS is equal to the ratio of heat ΔQ reversibly exchanged with the environment and the thermodynamic temperature *T* at which this exchange occurs, viz., ΔS=ΔQ/T. If the heat is absorbed from the environment by the thermodynamic system, ΔQ>0, leading to ΔS>0, while if it is released by the system into the environment, ΔQ<0. Here, Δ indicates the finite variation experienced either by entropy or by heat when the thermodynamic system passes from its initial to its final state supposed in thermodynamic equilibrium. The relationship between entropy and heat energy in terms of their variations can be regarded as the quantitative formulation of the Second Law of Thermodynamics based on a classical framework [4,5,6,7,8,9,10]. This relationship has been given by Clausius also in differential form, and for reversible processes, it is dS=δQ/T, where dS is the infinitesimal entropy change due to the infinitesimal heat δQ reversibly exchanged. It should be emphasized that while δQ is not an exact differential, dS is a complete (exact) differential due to the integrating factor 1/T, so *S* is a function of state depending only on the initial and final thermodynamic states.

The so-called equivalent formulation of the Second Law of Thermodynamics is represented by the Kelvin Statement, according to which it is not possible to convert all the heat absorbed from an external reservoir at higher temperature into work: part of it will be transferred to an external reservoir at lower temperature. Specifically, the proof of the equivalence between the Kelvin Statement and Clausius’ First Statement is outlined in [11]. The foundations of the Kelvin Statement were laid in [12,13], where the consequences of the Carnot proposition, according to which there is a waste of mechanical energy when heat is allowed to pass from one body to another at lower temperature, are discussed. This statement has been historically referred to as the Kelvin–Planck Statement [11,14]. However, another pioneer of this principle was Ostwald, who completed the Kelvin Statement by formulating the *Perpetuum Mobile* of the Second Kind. For this reason, in this paper, we combine together the two pioneering contributions, and we refer to the Kelvin–Ostwald Statement. Planck’s role in pioneering this principle may not have been major but was mainly focused in its divulgation within the scientific community. Note that recently, Moran et al. [15] have discussed the Second Law of Thermodynamics by using Clausius’ First Statement and the Kelvin Statement, but they did not take into account Clausius’ Second Statement and the Carnot Type Statement.

In strict relation with the above-mentioned general statements, it can be stated that according to the Second Law of Thermodynamics, it is impossible to realize a *Perpetuum Mobile* of the Second Kind able to use the internal energy of only one heat reservoir (see also Section 1.2 for more details).

As a general note, the entity of entropy variations in the Second Law of Thermodynamics applied to isolated thermodynamic systems allows for the distinction between reversible processes, i.e., ideal processes for which ΔS=0 (dS=0), and irreversible processes, i.e., real processes for which ΔS>0 (dS>0). To study reversible and irreversible thermodynamic processes, it is not sufficient to consider the thermodynamic system itself, but it is also necessary to consider its local and nonlocal surroundings, defining together with the system what is called the thermodynamic universe [11,14]. It describes the constructive role of entropy growth and makes the case that energy matters but entropy growth matters more.

At a later time, the Second Law of Thermodynamics was reformulated mathematically and geometrically again in a classical thermodynamic framework, through the so-called Carathéodory’s Statement [16], which exists in different formulations appearing in textbooks and articles (see, e.g., [11,14]), even though they are very similar to one another. Following Koczan [17], Carathéodory’s Statement states that “In the surroundings of each thermodynamic state, there are states that cannot be achieved by adiabatic processes”. This statement has often been given without a rigorous proof of equivalence with other formulations, but it has also been proved, even though not so rigorously, that this statement is a direct consequence of Clausius’ and Kelvin Statements [18,19,20]. However, a recent critique of this statement [21] shows that it already received strong criticism by Planck, and the issue related to its necessity and validity is still a matter of some discussion.

Afterwards, the Second Law of Thermodynamics acquired more general significance when a statistical physics definition of entropy was introduced by Boltzmann and Planck [22,23,24,25,26] and then generalized by the same Boltzmann [27] and by Gibbs [28]—this aspect was not discussed by the authors in previous works [14,17]. The main advancement was the celebrated Boltzmann probabilistic formula proposed within his kinetic theory of gases: $S_{B}=k_{B}\ln W$, with $k_{B}$ being the Boltzmann constant and W the number of microstates which characterize a gas’s macrostate. This formula was put forward and interpreted later by Planck [29] and is also known as the Boltzmann–Planck relationship: the entropy of a thermodynamic system is proportional to the number of ways its atoms or molecules can be arranged in different thermodynamic microstates. If the occupation probabilities of every microstate are different, it can be shown that Boltzmann formula is written in the Gibbs form, or Boltzmann–Gibbs form, valid also for states far from thermodynamic equilibrium: $S_{G}=-k_{B}\sum \;p_{i}\ln p_{i}$, with $p_{i}$ being the probability that microstate *i* has energy $E_{i}$ during the system energy fluctuation. It is straightforward to show that the Boltzmann–Gibbs entropy infinitesimal change in a canonical ensemble is equivalent to the Clausius’ entropy change, and thanks to this equivalence, the Second Law of Thermodynamics is also formulated within a statistical physics approach.

It is important to point out that very recently, Koczan has shown that Clausius’ and Kelvin Statements are not exhaustive formulations of the Second Law of Thermodynamics if compared with the Carnot Type Statement [17]. Specifically, it has been proved that the Kelvin Statement is a weaker statement (or, more strictly, non-equivalent) than Clausius’ First Statement, and Clausius’ First Statement is a weaker statement than the Carnot Type Statement, which can be considered equivalent to Clausius’ Second Statement. By indicating the heat absorbed by a heat source at temperature $T_{1}$ with $Q_{1}$ ($Q_{1}>0$) and the work resulting from the conversion of heat with *W* in the device operating between the heat source and the heat receiver at temperature $T_{2}$, the Carnot Type Statement states that $W/Q_{1}\leq \left(T_{1}-T_{2}\right)/T_{1}$, with $W/Q_{1}$ being the efficiency η; the efficiency of the conversion process of heat $Q_{1}$ into work *W* in the device operating in the range between $T_{1}$ and $T_{2}$ cannot be greater than the ratio of the difference between the two temperatures and the temperature of the heat source. Koczan’s arguments [17] about Clausius’ and Kelvin Statements were presented by Gadomski [30], who reiterated that the Kelvin Statement and Clausius’ First Statement are not exhaustive formulations of the Second Law of Thermodynamics and that instead, the Carnot Type Statement is more far-reaching for such a formulation. In this respect, very recently, the concept of entropy and the Second Law of Thermodynamics were reinterpreted and reformulated by Neukart [31] to assess the complexity of computational problems, giving a thermodynamic perspective on computational complexity.

This article ignores Carathéodory’s Statement, showing that there is not a real equivalence among this statement and Clausius’ and Kelvin Statements of the Second Law of Thermodynamics, contrary to what is asserted in some textbooks (see, e.g., [11]). Instead, more attention is devoted to the problem of deriving the Second Law of Thermodynamics from statistical physics. The problem of reversibility and irreversibility related to the usual formulation of the Second Law of Thermodynamics in terms of increasing entropy is also revised. It turns out that it cannot be always considered a fundamental and elementary law of statistical physics and, in some cases, even not completely true, as recently proved via the fluctuation theorem [32,33]. At the same time, it is also shown that the Second Law of Thermodynamics is a phenomenological law of nature working extremely well for describing the behavior of several thermodynamic systems. In this respect, one of the aims of this work is to explain the above-mentioned discordance based on precise and novel formulations of the Second Law of Thermodynamics.

More specifically, it is shown that three propositions can be formulated by taking into account that the thermodynamic formulations are non-exhaustive and the statistical formulations are incomplete and partly paradoxical: (i) the Kelvin–Ostwald Statement is strengthened by replacing *Perpetuum Mobile* of the Second Kind with the *Perpetuum Mobile* of the Third Kind by means of the Carnot Type Statement, according to which even this latter engine does not exist (Proposition 1); (ii) the Inequality of Heat and Temperature Proportions are stronger than Clausius’ First Statement (Proposition 2); and (iii) according to the Probabilistic Scheme of the Second Law of Thermodynamics, the expected value of the change in the elementary entropy in the state of non-maximum probability beyond thermodynamic equilibrium is positive (Proposition 3). Moreover, the Refrigeration *Perpetuum Mobile*, which is a practically unknown (or somewhat forgotten) limitation for heat pumps and refrigerators, is introduced.

This paper is organized as follows: Section 1 (introductory review chapter) presents the basic definitions and statements of phenomenological thermodynamics and statistical mechanics, including the various definitions of entropy, Boltzmann’s *H* theorem and the fluctuation theorems. Section 2 is devoted to some general clarifications of the Second Law of Thermodynamics in its purely phenomenological and classical form, while Section 3 deals with some important clarifications of this law in statistical physics. These three main sections are divided into four detailed subsections. Conclusions are drawn in Section 4.

### 1.2. Basic Definitions of Phenomenological Thermodynamics

A very important concept that led to the formulation of the laws of thermodynamics was the *Perpetuum Mobile*. However, even in the classical sense, there are several types of it. It is, therefore, worth clarifying this type terminology now, so that it can be further developed and used effectively in later sections of the article.

The *Perpetuum Mobile* is most often understood to mean a hypothetical device which would operate contrary to the accepted laws of physics. Usually, by *Perpetuum Mobile*, we mean a heat engine or a heat machine; therefore, it has the following basic definitions.

**Definition 1** (*Perpetuum Mobile* of Kind Zero and *Perpetuum Mobile* of the First Kind)**.**
*The “Perpetuum Mobile” of Kind Zero is a hypothetical device which moves forever without power and appears to be free from resistance to motion. This device neither performs work nor emits heat. On the other hand, a “Perpetuum Mobile” of the First Kind is a hypothetical device which performs work without an external energy source (it has infinite efficiency or greater than 100%).*


Sometimes, the *Perpetuum Mobile* of Kind Zero is referred to as the Third Kind. However, number 0 better reflects the original Latin meaning of the word (from Latin *perpetuum mobile* = perpetual motion), and number III will be used further. Anyway, in traditional thermodynamics, the greatest emphasis, not necessarily rightly, was placed on number II.

**Definition 2** (*Perpetuum Mobile* of the Second Kind)**.**
*The “Perpetuum Mobile” of the Second Kind is a hypothetical lossless operating warm engine with an efficiency of exactly 100%.*


It is quite obvious that the Second and Third Kinds do not exist, but it is not clear that Kind Zero cannot exist. For example, the solar system, which has existed in an unchanged form for millions of years, seems to be of Kind Zero in kinematic terms. Of course, we assume here that the energy of the Sun’s thermonuclear processes does not directly affect the motion of the planets.

Since the First Law of Thermodynamics was recognized historically later than the second one, we will introduce a special form of the First Law for the needs of the Second Law of Thermodynamics.

**Definition 3** (First Law of Thermodynamics for Two Heat Reservoirs)**.**
*We consider thermal processes taking place in contact with two reservoirs: a radiator with a temperature $T_{1}$ and a cooler with a temperature $T_{2}<T_{1}$. We denote the heat released by the radiator by $Q_{1}$ and the heat absorbed by the cooler by $Q_{2}$. We assume that we can only transfer work W≥0 to the environment or absorb work W≤0 but we cannot remove or absorb heat from the environment. Therefore, the principle of conservation of energy for the process takes the form*

(1)
$$Q_{1}=Q_{2}+W,$$

*which can be written, following the example of the First Law of Thermodynamics, as an expression for the change in the internal energy E of the system:*

(2)
$$\Delta E=-Q_{1}+Q_{2}=-W.$$

*In the set of processes, all possible signs and zeros are allowed for the quantities $Q_{1},Q_{2}$ and W that meet condition (Equation 1) or (Equation 2). In addition, thermodynamic processes can be combined (added) but not necessarily subtracted.*


The specific form of the First Law of Thermodynamics for a system of two reservoirs results from the assumption of internal heat exchange only. The definition of the First Law of Thermodynamics formulated in this way creates a space of energetically allowed processes for the Second Law of Thermodynamics. However, the Second Law of Thermodynamics introduces certain restrictions on these processes (see [17]).

**Definition 4** (Clausius’ First Statement)**.**
*Heat naturally flows from a body at higher temperature to a body at lower temperature. Therefore, a direct (not forced by work) process of heat transfer from the body at lower temperature to the body at higher temperature is not possible. Clausius’ First Statement allows for a large set of possible physical processes which do not violate the First Law of Thermodynamics.*


**Definition 5** (Kelvin–Ostwald Statement)**.**
*The processes of converting heat into work and work into heat do not run symmetrically. A full conversion of work into heat (internal energy) is possible. However, a full conversion of heat into work is not possible in a cyclical process. In other words, there is no Perpetuum Mobile of the Second Kind. The Kelvin–Ostwald Principle allows for the existence of a large set of possible physical processes which do not create a Perpetuum Mobile of the Second Kind and do not violate the First Law of Thermodynamics.*


To provide a more comprehensive statement of the Second Law of Thermodynamics, thermodynamic entropy must be defined. It turns out that this can be performed in thermodynamics in three subtly different ways. We will see further that in statistical mechanics, contrary to appearances, there are even more possibilities.

**Definition 6** (C-type Entropy for Reservoirs)**.**
*There is a thermodynamic function of the state of a system of two heat reservoirs. Its change is equal to the sum of the ratios of heat absorbed by these reservoirs to the absolute temperature of these reservoirs:*




(3)
$$\Delta S_{C}=\Delta S_{C1}+\Delta S_{C2}=\frac{-Q_{1}}{T_{1}}+\frac{Q_{2}}{T_{2}}.$$

*The minus sign results from the convention that a reservoir with higher temperature $T_{1}$ releases heat (for $Q_{1}>0$). However, if $Q_{1}<0$, then by the same convention, the reservoir with temperature $T_{1}$ will absorb heat.*


In this version, the total entropy change in the heat reservoirs is considered. We assume that the devices operating between these reservoirs operate cyclically, so their entropy remains unchanged. The definition of C-type entropy assumes that the heat capacity of the reservoirs is so large that the heat flow does not change their temperatures. Following Clausius, we can of course generalize the definition of entropy to processes with changing temperature.

**Definition 7** (*Q*- and *q*-type Clausius Entropies)**.**
*The entropy change in a thermodynamic system or part of it is the integral of the heat gain divided by the temperature of that system or part of it, i.e.,*

(4)
$$\Delta S_{Q}=\int \frac{\delta Q}{T},$$

*whereby if we use the temperature of the surroundings instead of the system temperature, we define the original (somewhat forgotten) Clausius’ entropy:*

$$\Delta S_{q}=\int \frac{\delta Q}{T_{surr}}.$$



A simple analysis based on Clausius’ First Statement leads to entropy inequalities of *Q*- and *q*-types:$$\Delta S_{Q}\geq \Delta S_{q},$$
where the equality can only apply to isothermal or trivial adiabatic processes (0 = 0). Applying this inequality to cyclic processes ($\Delta S_{Q}=0$) leads to Clausius’ inequality ($\Delta S_{q}\leq 0$), which became the basis of the entropic formulation of the Second Law of Thermodynamics. Of course, the change in the sign of this inequality resulted from further redefinitions of the entropy of the *q*-type to the entropy of the *C*-type and similar twists.

The *Q*-type entropy formula can apply not only to reversible processes but also to some irreversible ones. The simplest example is the equalization of temperatures of two bodies. It turns out that in some other irreversible processes, entropy increases despite the lack of heat flow. Therefore, we can further generalize the entropy formula, e.g., for the expansion of gas into vacuum [34].

**Definition 8** (*V*-type Entropy)**.**
*The entropy change in a gas is equal to the integral of the sum of the increase in internal energy and the product of the pressure and the increase in volume divided by the temperature of the gas:*

(5)
$$\Delta S_{V}=\int \frac{dE+pdV}{T}.$$



Entropies of the *V*- and *Q*-types seem apparently equal, but it will be shown later that this need not be the case—which also applies to entropy of the C-type. However, for any entropy (by default of the C- or *Q*-type), the following is considered the most popular Statement of the Second Law of Thermodynamics (see [17]).

**Definition 9.** (Clausius’ Second Statement)**.**
*There is a function of the state of a thermodynamic system called entropy, which reaches a maximum when the system is in a state of thermodynamic equilibrium. In other words, in thermally insulated systems, the only possible cyclical processes are those in which the total entropy of the system (two heat reservoirs) does not decrease:*

(6)
$$\Delta S_{tot}=\Delta S_{C1}+\Delta S_{C2}+\Delta S_{Q}=\Delta S_{C}\geq 0\;\;\;\;\left(\Delta S_{Q}=0\;\;for\;\;gaz\right).$$

*For reversible processes, the increase in total entropy is zero, and for irreversible processes, it is greater than zero. The same statement can also be applied to the Q-type entropy used for reservoirs ($\Delta S_{Qr}=\Delta S_{C}$). In the language of the q-type entropy for a gas ($\Delta S_{q}=-\Delta S_{C}$), the statement is Clausius’ inequality:*

$$-\Delta S_{q}\geq 0.$$



Assuming a given definition of entropy and thanks to the principle of its additivity, Clausius’ Second Statement is a direct condition for every process (possible and impossible). Therefore, this statement immediately states which process is possible and which is impossible and should be rejected according to this statement.

However, Clausius’ First Statement and the Kelvin–Ostwald Statement allow us to reject only some of the impossible processes. We reject the remaining part of the impossible processes on the basis of adding processes leading to the impossible ones. Possible processes are those which, as a result of arbitrary addition, do not lead to impossible processes. It has been shown that Clausius’ Second Statement is stronger and not equivalent to Clausius’ First Statement and the Kelvin–Ostwald Statement [17].

### 1.3. A Review of Statistical Definitions of Entropy and Its Application to an Ideal Gas

It is also worth considering statistical definitions of entropy, a topic not discussed in [14,17]. Contrary to appearances, this issue is not disambiguated in relation to phenomenological thermodynamics. On the contrary, there are many different aspects and cases of entropy in statistical physics. It is probably not even possible to talk about a strict universal definition, but only about definitions for characteristic types of statistical systems. Most often, definitions of entropy in statistical physics refer *implicitly* to an equilibrium situation—that is, a state with the maximum possible entropy. From the point of view of the analysis of the Second Law of Thermodynamics, this situation seems loopy—we postulate entropy maximization, but we define maximum entropy. However, dividing the system into subsystems makes sense of such a procedure. This shows a sample of the problem of defining entropy and the problem of precisely formulating the Second Law of Thermodynamics. Boltzmann’s general definition is considered to be the first historical static definition of entropy.

**Definition 10** (General Boltzmann Entropy for Number of Microstates)**.**
*The entropy of the state of a macroscopic system is a logarithmic measure of the number of microstates W that can realize the given macrostate, assuming a small but finite resolution of distinguishing microscopic states (in terms of location in the volume and values of speed, energy or temperature):*

(7)
$$S_{B}:=k_{B}\ln W,$$

*where the proportionality coefficient is the Boltzmann constant, $k_{B}$. The counting of states in classical mechanics is arbitrary, in the sense that it depends on the sizes of the elementary resolution cells. However, due to the property of the logarithm function for large numbers, the resolution parameters should only affect the entropy value in an additive way. In the framework of quantum mechanics, counting states is more literal, but for example, for an ideal gas without rotational degrees of freedom, it should lead to essentially the same entropy value.*


However, entropy defined in this way cannot be completely unambiguous as to the additive constant. Even in quantum mechanics, not all microstates are completely quantized, so there may be an element of arbitrariness in the choice of resolution parameters. Even if the quantum mechanics method for an ideal gas were unambiguous, it should be realized that at low temperatures, such a method loses its physical sense and cannot be consistent with the experiment. Therefore, to eliminate the freedom of constant additive entropy, the Third Law of Thermodynamics is introduced, which postulates the zeroing of entropy as the temperature approaches zero. However, such a principle has only a conventional and definitional character and cannot be treated as a fundamental solution—and should not even be treated as a law of physics. Moreover, it does not apply to an ideal gas, because no constant will eliminate the logarithm singularity at zero temperature.

Sometimes, in the definition of Boltzmann entropy, other symbols are used instead of W: Ω,|Ω| or ω. However, the use of the Greek letter omega may be misleading, because it suggests a reference to all microstates of the system, not just those that realize a given macrostate. Therefore, in this article, it was decided to use the letter W, just like on Boltzmann’s tombstone—but in a decorative “mathcal” version to distinguish it from the work symbol *W*.

Until we specify what microstates we consider to realize a given macrostate, (i) we do not even know whether we can count the microstates of the non-equilibrium type in the entropy formula. Similarly, the following is not clear: (ii) can the Boltzmann definition apply to non-equilibrium macrostates? Regarding problem (ii), it seems that the Boltzmann definition can be used for non-equilibrium states, even though the equilibrium entropy formulas are most often given. The latter results from the simple fact that the equilibrium state is described by fewer parameters. For example, when specifying the parameters of the gas state, we mean (consciously or not) the equilibrium state. Therefore, consistently regarding (i), if we have a non-equilibrium macrostate, then its entropy must be calculated after the non-equilibrium microstates. If the macrostate is in equilibrium, it is logical to calculate the entropy after the microstates relating to equilibrium. Unfortunately, some definitions of entropy force the counting of non-equilibrium states. This is the case, for example, in the microcanonical ensemble, in which one should consider a state in which one particle has taken over the energy of the entire system. Fortunately, this condition has negligible importance (probability), so problem (i) is not critical. This is because probability, the Second Law of Thermodynamics and entropy distinguish equilibrium states. However, this distinction is a result of the nature of things and should not be put into place by hand at the level of definition.

It is worth making the definition of Boltzmann entropy a bit more specific. Let us consider a large $N\geq N_{A}$ set of identical particles (but not necessarily quantum-indistinguishable). Let us assume that we recognize a given macrostate as a specific filling of *k* cells into which the phase space has been divided (the space of positions and momenta of one particle—not to be confused with the full configuration space of all particles). Each cell contains a certain number of particles, e.g., in the *i*th cell there are $n_{i}\geq 0$ particles. Of course, the number of particles must sum to *N*:(8)$$n_{1}+n_{2}+n_{3}+\ldots +n_{k}=N.$$ Now, the possible number of all configurations is equal to the number of permutations with repetitions, because we do not distinguish permutations within one cell:(9)$$W=\frac{N!}{n_{1}!\;n_{2}!\;n_{3}!\;\ldots \;n_{k}!}.$$

In mathematical considerations, one should be prepared for a formally infinite number of cells *k* and, consequently, fractional or even smaller-than-unity values of $n_{i}$. However, even in such cases, it is possible to calculate a finite number of states W. Taking the above into account, let us define Boltzmann entropy in relation to the Maxwell–Boltzmann distribution.

**Definition 11** (Boltzmann Entropy at a Specific Temperature)**.**
*The entropy of an equilibrium macroscopic system with volume V and temperature T, composed of N “point” particles, is a logarithmic measure of the number of microstates W that can realize this homogeneous macrostate by assuming small-volume cells in the space of positions υ and the space of momentum μ:*

(10)
$$S_{BT}:=k_{B}\ln W_{\upsilon ,\mu }\left(N,V,T\right),$$

*where the proportionality coefficient is the Boltzmann constant, $k_{B}$. The counting of states within classical mechanics is arbitrary here, in the sense that it depends on the volume υ,μ of the unit cells. However, thanks to the property of the logarithm function for large numbers, these parameters only affect the additive entropy constant (and also allow the arguments of the logarithmic function to be written in dimensionless form).*


An elementary formula for this type of Boltzmann entropy can be derived by using formula (Equation 9) for the distribution in space cells and the Maxwell–Boltzmann distribution for momentum space (the temperature-dependent part) [35,36]:(11)$$S_{BT}\approx Nk_{B}\ln\frac{V}{\upsilon }+\frac{3}{2}Nk_{B}\ln\frac{T}{\tau }+\frac{3}{2}Nk_{B}+\mathrm{const}\left(N,\upsilon ,\mu \right),$$
where the cell size of the shot space of volume μ has been replaced for simplicity by the equivalent temperature “pixel”:(12)$$\tau =\frac{\mu^{2/3}}{3k_{B}m},$$
where *m* is the mass of one particle.

In formula (Equation 11), what is disturbing is the presence of an extensive variable *V* in the logarithm, instead of an intensive combination of variables V/N. In [35] (p. 376), and in [36] (pp. 72, 73), in entropy derivation, division by *N* does not appear explicitly or implicitly via an additive constant either. The question is whether it is an error of this type of derivation (in two sources) or an error of the definition, which should, for example, include some dividing factor of the Gibbs type. This issue is taken up further in this work. However, it is worth seeing that in the volume part, formula (Equation 11) for $2\;m^{3}$ of air gives an entropy value more than twice as large as for $1\;m^{3}$, and it should not be like that.

The second doubt concerns the coefficient 3/2 in the expression $Nk_{B}$ in the derivations in [35] (p. 378) and in [36] (p. 72), which also appears in the entropy derived by Sackur in 1913 (see [37]). In further alternative calculations, the ratio is 5/2—e.g., in Tetrode’s calculations from 1912 (see [37]). It is worth adding that we are considering here a gas of material “points” (so-called monatomic) with the number of degrees of freedom for a determinate particle of 3, not 5. It is, therefore, difficult to indicate the reason for the discrepancy and to determine whether it is important due to the frequent omission of the considered term (or terms conditioned by “random” constants). There is quite a popular erroneous opinion that the considered term can only be derived on the basis of statistical mechanics, and it cannot be performed within the framework of phenomenological thermodynamics. In other words, if, at a constant volume, we start to increase the number of particles at the same temperature, then, according to the definition of the *V*-type, we will obtain the entropy term under consideration with the coefficient 3/2. However, if the same were calculated formally at constant pressure, the coefficient would be 5/2.

A steady-temperature state subject to the Maxwell–Boltzmann distribution is effectively a canonical ensemble—this will be analyzed further in these terms. One can also consider a microcanonical ensemble in which the energy *E* of the system is fixed. This leads to a slightly different way of understanding Boltzmann entropy.

**Definition 12** (Boltzmann Entropy at a Specific Energy)**.**
*The entropy of a macroscopic system with volume V and energy E, composed of N particles, is a logarithmic measure of the number of microstates W that can realize this homogeneous macrostate by assuming small cells of volume υ and low resolution of energy levels ε:*

(13)
$$S_{BE}:=k_{B}\ln W_{\upsilon ,\varepsilon }\left(N,V,E\right),$$

*where the proportionality coefficient is the Boltzmann constant, $k_{B}$. The counting of states within classical mechanics is arbitrary in the sense that it depends on the choice of the size of the elementary volume cell and the choice of the width of the energy intervals. However, due to the property of the logarithm function for large numbers, the parameters υ and ε only affect the entropy value in an additive way.*


The calculation of this type of Boltzmann entropy for the part of the spatial volume is the same as before. Unfortunately, calculating the energy part is much more difficult. First, one must consider the partition of energy *E* into individual molecules. Secondly, the isotropic velocity distribution should be considered here without assuming the Maxwell–Boltzmann distribution. Therefore, the counting of states should take place over a larger set than for equilibrium states, but for states that are isotropic in terms of velocity distribution.

Counting states by using the partition function is rarely performed because it is cumbersome. An example is represented by the calculations for the photon sphere of a black hole in the context of Hawking radiation [34]. These calculations led to a result consistent with the equation of state of the photon gas, but compliance with Hawking radiation was obtained only for low energies.

While the Boltzmann entropy of the $S_{BT}$-type was defined directly for equilibrium states, and the entropy of the $S_{BE}$-type counted isotropic but not necessarily equilibrium states, one can also consider the entropy counting even non-isotropic states. This takes place in the extended phase space, i.e., in the configuration space, where the state of the microstate is described by one point in the 6N-dimensional space. In other words, there is another approach to measuring the complexity of a macrostate from previous ones. Instead of dividing the space of positions and of momentum into cells and counting all possibilities, we can take, as a measure, the volume of configurational phase space that a given macrostate could realize. Unfortunately, energy resolution or cell size is also somewhat useful here.

**Definition 13** (Boltzmann Entropy for a Phase Volume with the Gibbs Divider)**.**
*The entropy of a macroscopic system with a volume V, composed of N particles, is a logarithmic measure of the phase volume $\bar{\Gamma }\cdot \varepsilon$ of all microstates of the system with energy in the range (E,E+ε), which can realize a given macrostate with energy in this range:*

(14)
$$S_{\Gamma }:=k_{B}\ln\frac{\bar{\Gamma }\left(N,V,E\right)\cdot \varepsilon }{N!\;\omega }=:k_{B}\ln\frac{W_{\Gamma }}{N!},$$

*where in addition to the unit volume of the ω phase-space configuration cell, there is the so-called Gibbs divider N!. The role of the Gibbs divider is to reduce a significant value of the phase volume and is sometimes interpreted on the basis of the indistinguishability of identical particles. The dependence on the parameters ε and ω is additive.*


The definition could be limited to the surface area Γ of the hypersphere in the configurational phase space instead of the volume $\bar{\Gamma }\cdot \varepsilon$ of the hypersphere shell. This would remove the parameter ε but would formally require changing the volume of the unit cell to a unit area one dimension smaller.

In Boltzmann phase entropy, counting takes place over all microstates from the considered energy range. Therefore, both non-isotropic states in velocity and non-uniform states in position—in short, non-equilibrium states—are taken into account. Nevertheless, we will refer the entropy to the equilibrium state, since no non-equilibrium parameters are given for the state. The entropy of non-equilibrium macrostates can be found by dividing them into subsystems that can be treated as equilibrium and then adding their entropies.

It is sometimes suggested e.g., see [36] (p. 97) that the volume of a unit cell in phase space follows from the Heisenberg uncertainty principle, the strict form of which is $\sigma_{x}\sigma_{p_{x}}\geq \hbar /2=h/\left(4\pi \right)$. Then, it should be $\omega ={\left(\hbar /2\right)}^{3N}$, although it is most often defined more simply as $\omega :=h^{3N}$. The units are more important here than the values themselves, but the problem of the lack of a divisor 4π is rather decidable in quantum computations. This can be seen most easily from the de Broglie relation pλ=h, which defines the volume of two-dimensional phase space (for momentum and position in the range of the matter wavelength λ). A more detailed calculation for a single particle involves quantizing the momentum on a cubic L×L×L box. In the quasiclassical limit, the summation over discrete states turns into a continuous integral over phase space see [38] (pp. 181, 415):$$\sum_{\vec{p}\in \frac{h}{L}Z^{3}}⟶\;\;\frac{L^{3}}{h^{3}}\int d^{3}p=\frac{1}{h^{3}}\int d^{3}r\int d^{3}p.$$ So, there is indeed a divisor $h^{3}$ here, which for *N* particles will be $h^{3N}$. The limit is quasiclassical because if the size of the unit cell tends to zero (h→0), the entropy would tend to infinity. Therefore, quantum mechanics is believed to specify a finite value of entropy even at the level of the additive constant. It is worth knowing, however, that this is a theoretical assumption that is not necessary for thermodynamic or even statistical considerations. In other words, most of such considerations do not even depend on the arbitrary choices of the additive entropy constant and the size of the elementary cells.

The area of the constant energy phase hypersurface of dimensions 6N−1 can be calculated from the exact formula for the area of an odd-dimensional hypersphere of radius *r* immersed in even-dimensional space:(15)$$A_{6N-1}=V^{N}A_{3N-1}\left(r\right),$$
(16)$$A_{3N-1}\left(r\right)=\frac{2\pi^{3N/2}}{\Gamma (3N/2)}r^{3N-1}.$$ Taking into account that the radius of the sphere in the configurational momentum space can be related to the energy for material points $r=p=\sqrt{2mE}$ as follows, we obtain
(17)$$\frac{W_{\Gamma }}{N!}=\frac{\bar{\Gamma }\left(N,V,E\right)\cdot \varepsilon }{N!\;\omega }=\frac{V^{N}2\pi^{3N/2}\;\varepsilon }{N!\;\Gamma \left(3N/2\right)\;h^{3N}}{\sqrt{2mE}}^{3N-1}.$$ For further simplifications, Stirling’s asymptotic formula is standardly used:(18)$$\Gamma \left(n\right)=\left(n-1\right)!≅\sqrt{2\pi n}{\left(\frac{n}{e}\right)}^{n}\approx n!.$$ Its application to further approximations leads to
(19)lnN!≈NlnN−N,
(20)$$\ln\Gamma \left(3N/2\right)\approx \frac{3}{2}N\ln N-\frac{3}{2}N.$$ On this basis, the Boltzmann phase entropy of an ideal gas takes the form
(21)$$S_{\Gamma }\approx Nk_{B}\ln\frac{V}{N\upsilon }+\frac{3}{2}Nk_{B}\ln\frac{E}{N\varepsilon }+\frac{5}{2}Nk_{B}+\;\mathrm{const}\left(N,m,\varepsilon ,h\right),$$
where the auxiliary dimensional constant υ was chosen so that the volume of the phase cell of one particle is $\upsilon {\sqrt{2m\varepsilon }}^{3}=h^{3}$. It is customary to simply omit the dimensional constants and the additive constant (but not in the so-called Sackur–Tetrode entropy formula). In any case, here, the simplest part of the additive term has a coefficient of 5/2, and not 3/2 as before. Furthermore, the volume under the logarithm sign is divided by the number of particles *N*. Generally, the presented result and the derivation method are consistent with the Tetrode method (see [37]).

There is no temperature yet in the entropy formula obtained from the microcanonical decomposition. However, the general formalism of thermodynamics allows us to introduce temperature in a formal way:(22)$$\frac{1}{T}:={\left(\frac{\partial S}{\partial E}\right)}_{V,N}\approx \frac{3}{2}Nk_{B}\frac{1}{E}.$$ The formula uses the Boltzmann phase $S_{\Gamma }$ entropy, although the formula also applies to the $S_{BE}$ entropy; the later, however, has not been calculated (at least here). In any case, by using the above relation between temperature and energy, the $S_{\Gamma }$ entropy can be given in a form almost equivalent to the $S_{BT}$ entropy. The difference concerns the discussed factor dividing the volume and the less important term by the factors 5/2 vs. 3/2.

There is yet a slightly different approach to the statistical definition of entropy than Boltzmann’s original approach. Most often, it refers to the canonical distribution and is attributed to Gibbs. Gibbs was probably the first to use the new formula for statistical entropy, but Boltzmann did not shy away from this formula either (see below). In addition, Shannon used this formula in information theory, as well as in ordinary mathematical statistics without a physical context.

**Definition 14** (Gibbs or Gibbs–Shannon entropy)**.**
*The Gibbs entropy of a macroscopic system whose microscopic energy realizations have probabilities $p_{i}$ is given by*

(23)
$$S_{G}=:-k_{B}\sum_{i}p_{i}\ln p_{i},$$

*where the proportionality coefficient is the Boltzmann constant $k_{B}$ taken with a minus sign. The counting of states may be arbitrary, in the sense that they can be divided into the micro-scale realizations of a macrostate and, consequently, assigned a probability distribution. This entropy is sometimes also called Gibbs–Shannon entropy.*


The Gibbs entropy usually applies to a canonical system in which the energy of the system is not strictly defined, but the temperature is specified. Therefore, we can only talk about the dependence of entropy on the average energy value. In a sense, entropy (like energy) in the canonical distribution also has a secondary, resulting role—as a derivative of another quantity or as an average value. A more direct role is played by the Helmholtz free energy, which in thermodynamics is defined as follows:(24)F=〈E〉−TS.

Only from this energy, entropy can be calculated. This is an entropy equivalent to the Gibbs entropy, but due to a different way of obtaining it, we will treat it as a new definition.

**Definition 15** (Entropy of the *F*-type)**.**
*The entropy of a (canonical) system, expressed in terms of the Helmholtz free energy F, is, with opposite sign, equal to the partial derivative of this free energy with respect to temperature at fixed volume V and number of particles N:*

(25)
$$S_{F}:=-{\left(\frac{\partial F}{\partial T}\right)}_{V,N}.$$

*In statistical terms, the free energy F is proportional with opposite sign to the absolute temperature and to the logarithm of the statistical sum:*

(26)
$$F:=-k_{B}T\ln Z.$$

*The statistical sum is the normalization coefficient of the unnormalized exponential probabilities $P_{i}=p_{i}Z$ of energy in the thermal (canonical) distribution and is called partition function:*

(27)
$$Z:=\sum_{i}P_{i}:=\sum_{i}\exp\left(-\frac{E_{i}}{k_{B}T}\right).$$



Although this definition of entropy partially resembles the Boltzmann definition (e.g., instead of the number of states W, there is a statistical sum *Z*), it is more complex and specific (it includes temperature, which complicates the partial derivative). At least superficially, this entropy seems to be slightly different from the general idea of Boltzmann entropy (or even from Gibbs entropy), as
(28)$$S_{F}=k_{B}\ln Z+k_{B}T{\left(\frac{\partial \ln Z}{\partial T}\right)}_{V,N}=k_{B}\ln Z+\frac{1}{T}\langle E\rangle ,$$
but immediately, one sees that the additional last term contains only a part proportional to the number of particles *N* and is often omitted in many types of entropy (it does not apply to the Sackur–Tetrode entropy). However, in this work, it was decided to check the main part of this term, omitting the scale constants (Planck’s constant, the sizes of unit cells, the mass of particles and the logarithms of π and *e*).

For an ideal gas, the statistical sum *Z* can be calculated similarly to the calculation of the $S_{BT}$ entropy. For the calculation of the Boltzmann $S_{BT}$ entropy, we referred to arbitrary cells of the position and momentum space. Due to the discrete nature of the statistical sum, its calculation is usually performed as part of quantization on a cubic or cuboid box. The spatial boundary conditions on the box quantize momentum and energy, so the counting takes place only in terms of momentum quantum numbers—the volume of the box appears only indirectly in these relations. The result of this calculation is [39]
(29)$$Z=\frac{V^{N}}{N!\;h^{3N}}{\sqrt{2\pi mk_{B}T}}^{3N}.$$ One sees that this value is very similar to $W_{\Gamma }/N!$ when calculating the Boltzmann entropy of the $S_{\Gamma }$-type. Indeed, after making standard approximations, the first principal term ($k_{B}\ln Z$) of the entropy of an ideal gas of the $S_{F}$-type differs from the previous entropy only by a term proportional to the variable *N* itself. However, the additional term of this entropy will be $3Nk_{B}/2$ and will reconcile these entropies:(30)$$S_{F}\approx Nk_{B}\ln\frac{V}{N\upsilon }+\frac{3}{2}Nk_{B}\ln\frac{T}{\tau }+\frac{5}{2}Nk_{B}+\mathrm{const}\left(N,m,h,k_{B}\right),$$
where in this case, the auxiliary dimensional constants υ and τ satisfy the relation $\upsilon {\sqrt{2\pi mk_{B}\tau }}^{3}=h^{3}$.

In addition to the microcanonical and canonical (thermal) systems, there is also a large canonical system (grand canonical system), in which even the number of particles is not constant. Due to the numerous complications in defining entropy so far, the grand canonical system will not be considered here. Even more so, isobaric systems (isothermal or isenthalpic) will be omitted.

Note that the probability appearing in the Gibbs entropy is simply normalized to unity, and not to the number of particles *N*, which may suggest that the Gibbs entropy differs from the Boltzmann-type entropy in the absence of this multiplicative factor *N*. Indeed, this is reflected in the following version of the Boltzmann entropy, the Boltzmann entropy of the *H*-type, in which there is a single-particle probability distribution function in the phase space [40] normalized to *N* (and not to unity).

**Definition 16** (Boltzmann entropy of *H*-type)**.**
*In the state described in the phase space by the single-particle distribution function $f(t,\vec{r},\vec{v})$, with t being the time, $\vec{r}$ the spatial coordinate and $\vec{v}$ the velocity, the H-type entropy of the system (or simply the −H function, but not the Hamiltonian and not enthalpy) is defined by the following integral:*

(31)
$$S_{H}\left(t\right):=-k_{B}\int d^{3}r\int d^{3}v\;f\left(t,\vec{r},\vec{v}\right)\ln\frac{f(t,\vec{r},\vec{v})}{\varphi \;e}=:-k_{B}H\left(t\right),$$

*where the distribution function normalized to the number of particles N and φe in the denominator is the product of constants. The φ constant is usually the neglected volume of the elementary phase cell for positions and velocities (dimensional constant), and e is a useful numerical constant here.*


Note that the divisor *e*, after taking into account the normalization condition of the distribution function, leads to an additive entropy term of $Nk_{B}$ without the factor 3/2 or 5/2. However, perhaps, this divisor actually corrects the target entropy value.

Moreover, it is often assumed that the quantity defined above (or slightly modified) is not entropy but, taken with opposite sign, the so-called *H* function. However, in light of the various definitions cited in this work, there is no point in considering this quantity to be something different from entropy, all the more so because Boltzmann formulated a theorem regarding this function, which was supposed to reflect the Second Law of Thermodynamics. To formulate this theorem, the Boltzmann kinetic equation and the assumption of molecular chaos are also needed.

The Boltzmann kinetic equation for the one-particle distribution function $f(t,\vec{r},\vec{v})$ for a state of *N* particles (and normalized to *N*) postulates that the total derivative of the distribution function is equal to the partial derivative taking into account two-particle collisions (without which the total derivative of the distribution function would be zero, i.e., the Liouville equation):(32)$$\frac{df}{dt}:=\frac{\partial f}{\partial t}+\vec{v}\cdot \frac{\partial f}{\partial \vec{r}}+\vec{g}\cdot \frac{\partial f}{\partial \vec{v}}={\left(\frac{\partial f}{\partial t}\right)}_{coll},$$
where $\vec{g}$ is the external acceleration field—e.g., the gravitational field—and the subscript *coll* stands for collisions. The assumption of molecular chaos (*Stosszahlansatz*), in simple terms, consists in assuming that two-particle collisions are factored by using the one-particle distribution function as follows:(33)$${\left(\frac{\partial f}{\partial t}\right)}_{coll}\left(t,\vec{r},\vec{v}\right)=\int d\Omega \int d^{3}v_{1}\;\left|{\vec{v}}_{1}-\vec{v}\right|\left(f\left(t,\vec{r},{\vec{v}}_{1}^{\prime }\right)f\left(t,\vec{r},{\vec{v}}^{\prime }\right)-f\left(t,\vec{r},{\vec{v}}_{1}\right)f\left(t,\vec{r},\vec{v}\right)\right)\frac{d\sigma }{d\Omega },$$
where dσ/dΩ is the differential cross-section related to the velocity angles satisfying the relation $|{\vec{v}}_{1}-\vec{v}\left|=\right|{\vec{v}}_{1}^{\prime }-{\vec{v}}^{\prime }|$. The kinematic–geometric idea of the molecular chaos assumption is quite simple (it is a simplification), but the mathematical formula itself is already complex; therefore, we will not go into details here. Readers interested in the details can be referred to the extensive textbook [41], which describes an even richer set of equations, there called the Bogoliubov equations. In another source, this set of equations is called the BBGKY hierarchy (after Bogoliubov, Born, Green, Kirkwood and Yvon, respectively) [42].

It turned out that the assumption *Stosszahlansatz* together with evolution Equation (Equation 32) was enough for Boltzmann, in a sense, to derive the Second Law of Thermodynamics.

**Theorem 1** (Boltzmann *H*)**.**
*If the single-particle distribution function $f(t,\vec{r},\vec{v})$ satisfies the “Stosszahlansatz” assumption regarding Boltzmann evolution Equation (Equation 32), then the entropy of the Boltzmann H-type is a non-decreasing function of time:*

(34)
$$\frac{dS_{H}}{dt}\geq 0\;\;\;\;or\;\;\;\;\Delta S_{H}\geq 0.$$



The proof of Boltzmann’s theorem, in a notation analogous to the one adopted here, can be found in the textbooks [38,40,42].

Although Boltzmann’s theorem is a true mathematical theorem, it is unfortunately not (and cannot be) a derivation of the Second Law of Thermodynamics. The thesis of every theorem follows from an assumption, and the assumption of molecular chaos (*Stosszahlansatz*) is not entirely true. This assumption, in some strange way, introduces the irreversibility and asymmetries of time evolution into the macroscopic system, when the remaining equations of physics do not contain this element of time asymmetry. This issue has been called the irreversibility problem or even Loschmidt’s irreversibility paradox.

An interesting discussion on molecular chaos was undertaken by Huang, the author of the textbook [38] (pp. 85–91), who called the paradox of irreversibility a purely historical issue. Huang claims that molecular chaos is a local minimum of entropy (a local maximum of the *H* function), while the global maximum is the Maxwell–Boltzmann distribution (a global minimum of the *H* function). The evolution of the system does not proceed strictly according to the Boltzmann kinetic equation, but the entropy increases to a maximum value taking into account the fluctuation noise with local minima in states of molecular chaos and local maxima in other unidentified states. This increase in entropy is called the statistical approach to the Second Law of Thermodynamics. This image is quite suggestive, but a bit inconsistent and somewhat destructive in terms of knowing the evolution of the system. The presence of the destruction effect is clear and perhaps necessary—the strict evolution of the system was simply rejected in order to introduce local time reversal symmetry—without a rational justification that on average, the evolution proceeds according to the kinetic equation. The element of inconsistency concerns the suggestion (used in the proof of a local entropy minimum in a state of molecular chaos) that time reversal leads further to an increase in entropy (backwards in time). This leads to two ambiguities. First, it is not known whether this is how time reversal should work (perhaps it should). Second, even if the time reversal were correct, what would be the departure from Boltzmann kinetic evolution? Perhaps in the form of “backward time” evolution. But if so, how do we explain the advantage of forward-in-time evolution over backward-in-time evolution? In any case, this does not provide a new rigorous formulation of the Second Law of Thermodynamics, but only a somewhat speculative explanation of the reversibility paradox at the expense of Boltzmann kinetic evolution.

Typically, in the context of the irreversibility problem, it is claimed (somewhat incorrectly) that all the fundamental equations of physics are time-symmetric. However, this does not in any way apply to equations involving resistance to motion, including friction and viscosity. It is difficult to explain why the aspect of resistance to motion is so neglected in the history of physics. The ignorance of resistance to movement is imputed to Aristotle, when in fact it is exactly the opposite, and Aristotle described the proportion of motion that included resistance. This proportion of motion can even be interpreted as consistent with Newtonian dynamics according to the correspondence principle [43]. The time asymmetry of Aristotle’s equation (proportion) of dynamics was noticed by American physicist Leonard Susskind (see [43]).

Another example of an equation lacking time symmetry is the Langevin equation, which also relates to viscous friction and has applications in statistical mechanics ([36] p. 236). The anisotropic generalization of the Langevin equation is even used to describe the fission process of atomic nuclei [44]. If we generalize the Langevin equation for a point particle to a continuous medium (with viscosity), we obtain the Navier–Stokes equation (see [45] p. 45). Moreover, the general solutions of the Navier–Stokes equations constitute the oldest and still unsolved Millennium Prize Problem. This equation can be applied even to interstellar matter [46]. Another simpler example is the thermal conductivity equation, which almost by definition distinguishes the direction of time flow and therefore the direction of heat flow. One can only ask whether the Langevin, Navier–Stokes or heat conduction equations are fundamental equations. In a sense, all of these equations reflect the Second Law of Thermodynamics, the first and second ones implicitly and the third one explicitly in the form of Clausius’ First Statement. It is worth noting that the conduction equation is a first-order differential equation with respect to time. So, is it this property that determines the irreversibility of this equation in time? It seems that not only this is true but that it can also be compared with other equations with this property.

A good example of a general equation in statistical mechanics with a first derivative with respect to time is the Fokker–Planck equation (later also known as the Kolmogorov forward equation from 1931). This equation has applications even in nuclear physics [47]. The Fokker–Planck equation is a partial differential equation describing the evolution of the probability distribution function taking into account drift, diffusion and drag forces. In its most general version, the equation from Planck’s 1917 paper [48], the probability distribution is given in phase space, but most of the equations in that paper concern only position space. However, in the textbook [40] (pp. 91–93), this equation is presented only in momentum space as the space-averaged (integrated) Boltzmann kinetic equation in the diffusion approximation. At the same time, it was noted there that the form of the Fokker–Planck equation is so universal that it can be obtained in various variables. This is probably the reason why in the textbook [36], the Fokker–Planck equation on page 215 is a position equation, and on pages 236 and 238, it is already a phase equation (with velocity as a variable). Regardless of the subtleties mentioned, there is no doubt that the Fokker–Planck equation is not symmetric in time.

A very similar example is the Smoluchowski diffusion equation from 1916 in [49] (see also [50]). While the Fokker–Planck equation concerns the distribution function in position space or momentum space or both (phase space), the Smoluchowski diffusion equation concerns the distribution function in only position space in one dimension. In classical mechanics, the momentum and position images are not complementary, so it is difficult to compare the equations written in these two different images. However, the fact is that equation “(4)” from Smoluchowski’s 1916 paper [49] is a version of the Fokker–Planck equation that took its final form in Planck’s 1917 paper [48]. At the same time, for example, in Fokker’s work from 1914 [51], there is no such equation explicitly written down, nor even a substantially similar equation (with the second spatial derivative and the first time derivative—in explicit or Fourier form). The positional image allows for a more accurate description of Brownian motion, which Smoluchowski had already described earlier in 1906 with a formula now known as the Einstein–Smoluchowski relation [52].

In this respect, recently, it has been shown by Yuvan and Bier [53] that unlike diffusion systems, out-of-equilibrium statistical systems consisting of overdamped particles transporting or converting energy are not subjected to Gaussian noise anymore but to Lévy noise. The particles exhibit another type of Brownian motion generated by Lévy noise and thus obey to an α-stable nonhomogeneous distribution, or Lévy distribution. Bringing the systems back to thermodynamic equilibrium, the relaxation process leads to an increase in entropy which, in turn, causes a decrease in free energy.

The issue of the time derivative is of a completely different nature in quantum mechanics and for the Schrödinger equation, which was written only in 1926. This equation contains almost the same derivatives, including only the first derivative with respect to time, as the Fokker–Planck or Smoluchowski equation. Due to the complex nature of this equation and the imaginary unit in the time derivative, it is treated as invariant under time reversal. This statement requires the assumption of the complex conjugate accompanying time reversal.

In the context of Boltzmann and statistical physics, it is impossible not to mention the very fundamental ergodic hypothesis, which serves as a basic assumption and postulate. The ergodic hypothesis assumes that the average value of a physical quantity (random variable) based on a statistical distribution is realized over time, i.e., it tends to the average time value of this physical quantity (random variable). Let us assume that this pursuit simply boils down to the postulate of equality of these two types of averages (without precisely defining the period Δt of this averaging):(35)$$\langle X(t)\rangle \overset{!}{\approx }{\bar{X(t)}}_{\Delta t},$$
(36)$$\sum_{i}p_{i}\left(t\right)X_{i}\left(t\right)\overset{!}{\approx }\frac{1}{\Delta t}\int_{t}^{t+\Delta t}X\left(t^{\prime }\right)dt^{\prime }.$$ If the distribution is stationary (it does not evolve over time), then of course Δt can and should tend to infinity +∞. And this is indeed the standard assumption; however, averaging over the future or past or the entire timeline should not differ. However, for non-stationary (time-dependent) distributions, averaging over an infinite time seems to be pointless. In such situations, one could, therefore, consider “local” averages over time. Then, the forward time averaging used above could prefer to evolve forward in time relative to the symmetric averaging time interval. Unfortunately, ergodicity is limited to stationary situations, even where it supposedly does not always occur.

As one can see, the physical status of the ergodic hypothesis is not entirely clear. While the left side of the considered equalities has a clear statistical sense, it is not entirely clear how to determine the right side, which requires knowledge of the time evolution of the system under consideration. What extension of the Liouville equation should be used to describe this evolution in time, the Boltzmann equation or some other equivalent? In other words, it is not clear whether in practice, the ergodic hypothesis is a working definition of time averages (right-hand side) or a postulate or principle of physics that can be confirmed or disproved theoretically or experimentally. The literature states as an indisputable fact that given known systems are ergodic or not ergodic. Nevertheless, we should be more humble regarding the physical status of the ergodic hypothesis and treat it as an important methodological tool. In any case, it seems that the ergodic hypothesis should be independent of the Boltzmann equation with molecular chaos. Therefore, the problem of irreversibility cannot be solved merely by questioning the ergodic hypothesis. Ergodicity should not be confused with weak ergodicity, which is based on the Boltzmann equation as a simplifying assumption.

The violation of weak ergodicity (molecular chaos) probably has deeper reasons, since even a simple system of rigid spheres caused problems in the proof of this property (no violation) in the context of the Sinai theorem [54]. Anyway, more physical examples of magnetics, glasses (regular and spin) and colloids do not satisfy weak ergodicity or do not satisfy the molecular chaos hypothesis at the basis of the Boltzmann equation.

Regardless of the existence of resistance to motion, criticisms of the molecular chaos assumption, the problems with the status of ergodicity and Boltzmann’s generally enormous contribution, it seems that the essence of the Second Law of Thermodynamics in a statistical approach can be contained in the Newer Fluctuation Theorem and its corollary, the Inequality of the Second Law.

### 1.4. The Fluctuation Theorems and the Inequality of the Second Law of Thermodynamics

It is well established in classical thermodynamics that according to the well-known formulation of the Second Law of Thermodynamics, the entropy of an isolated thermodynamic system tends to increase until it reaches thermodynamic equilibrium. However, bearing in mind the statistical nature of entropy, an aspect not discussed in Ref. [14], recently, the here so-called Newer Fluctuation Theorem, which gives further information not contained in the Second Law of Thermodynamics, was formulated. This theorem states that there exists a relative probability density that the entropy production of an out-of-equilibrium thermodynamic system could fluctuate in time, viz., either $\Delta S_{t}>0$ or $\Delta S_{t}<0$: entropy can also decrease [32,33]. The notion of entropy production can be interpreted quasi-mechanically in the context of Clausius’ definition (*Q*- or *V*-type) [55].

**Definition 17** (Entropy Production)**.**
*In a system at temperature T, in which particles with a certain distribution of velocities $\vec{v}$ are subjected to a force $\vec{F}$ (of an optical nature [55]), the following entropy change occurs after time t:*

(37)
$$\Delta S_{t}:=\frac{1}{T}\int_{0}^{t}\vec{v}\left(t^{\prime }\right)\cdot \vec{F}\left(t^{\prime }\right)dt^{\prime },$$

*where we interpret the integral as some kind of mechanical work (equivalent to heat) and not as an averaging over time (we divide by T, not by t). Similarly, we also do not divide by the Boltzmann constant in order to preserve the natural physical unit of entropy.*


A newer version of the fluctuation theorem was formulated by Evans et al. [32] for non-equilibrium steady states that are thermostated by keeping constant the total energy. Afterwards, Gallavotti and Cohen proved the theorem for chaotic, isoenergetic non-equilibrium systems [56,57]. The fluctuation theorem in its newer version states the following.

**Theorem 2** (Newer Fluctuation Theorem)**.**
*In out-of-equilibrium thermodynamic systems, the ratio between the probability density that the entropy production $\Delta S_{t}$ takes the positive value $k_{B}\sigma$ and the probability density that it takes the negative value $-k_{B}\sigma$, in a time interval t, follows an exponential law and increases with the increase in σ:*

(38)
$$\frac{PD\left(\Delta S_{t}=k_{B}\sigma \right)}{PD\left(\Delta S_{t}=-k_{B}\sigma \right)}=\exp\left(\sigma \right),$$

*where $PD(\Delta S_{t}=k_{B}\sigma )$ or $PD(\Delta S_{t}=-k_{B}\sigma )$ is the probability density that $\Delta S_{t}$ assumes the value $k_{B}\sigma$ or $-k_{B}\sigma$, respectively, in a time interval t.*


The theorem was demonstrated via computer simulations, for example, by exploring the temporal evolution of the anisotropy in the probability of observing trajectory segments with positive average entropy production and their conjugate trajectory antisegments with negative average entropy production for reversible deterministic systems and for Hamiltonian systems with and without applied dissipative fields [58,59]. The theorem was also experimentally proved by observing the time-dependent relaxation of a colloidal particle under a step change in the strength of a stationary optical trap [60].

According to Equation (Equation 38), for a finite thermodynamic system in a time interval *t*, there is a non-zero probability density that the entropy production flows in a direction that is opposite to the one predicted by the Second Law of Thermodynamics. We emphasize that this theorem only apparently contradicts the Second Law of Thermodynamics. Indeed, the fluctuation theorem is valid for both microscopic and macroscopic systems, while the Second Law of Thermodynamics is valid only for macroscopic systems. The fluctuation theorem does prove the Second Law of Thermodynamics in its ordinary form, but it can provide it with a strict framework which will be explained in the following sections.

A consequence of the Newer Fluctuation Theorem is the Inequality of the Second Law of Thermodynamics expressed in terms of the ensemble average $\langle \Delta S_{t}\rangle$. The Inequality of the Second Law of Thermodynamics Theorem states the following.

**Theorem 3** (Inequality of the Second Law of Thermodynamics)**.**
*The ensemble average of entropy production $\langle \Delta S_{t}\rangle$ is not negative, for any non-negative time t, if the system was not in thermodynamic equilibrium at the initial time:*

(39)
$$\forall t\geq 0,\;S_{0}\neq S_{eq}:\;\;\;\;\langle \Delta S_{t}\rangle \geq 0,$$

*where $\langle \Delta S_{t}\rangle$ stands for the ensemble average of entropy production, function of the microstate of the thermodynamic system.*


Generally, the Theorem of Inequality of the Second Law of Thermodynamics is a corollary of the Newer Fluctuation Theorem, which represents an important step forward in statistical thermodynamics. In its derivation, there is no need to recur to the strong assumption of molecular chaos as performed by Boltzmann to prove the *H* theorem [22,23], namely, without assuming the velocities of particles uncorrelated and independent of position.

Note that there also exists an old version of the fluctuation theorem related to the relative fluctuations of the main physical quantities characterizing appropriate statistical ensembles. The Old Fluctuation Theorem can be formulated in the following form.

**Theorem 4** (Old Fluctuation Theorem)**.**
*In general, i.e., in appropriate statistical ensembles, the square of the relative fluctuations of energy, density and number of particles is inversely proportional to the average number of particles (somewhat similar for pressure):*

(40)
$$\frac{\langle E^{2}\rangle -{\langle E\rangle }^{2}}{{\langle E\rangle }^{2}}\approx \frac{\langle d^{2}\rangle -{\langle d\rangle }^{2}}{{\langle d\rangle }^{2}}\approx \frac{\langle N^{2}\rangle -{\langle N\rangle }^{2}}{{\langle N\rangle }^{2}}\approx \frac{1}{\langle N\rangle },\;\;\;\;\frac{\langle p^{2}\rangle -{\langle p\rangle }^{2}}{{\langle p\rangle }^{2}}\approx \frac{1}{{\langle N\rangle }^{2/3}}.$$



The formulation of the theses and the proof of this theorem can be found in Tolman’s book [61] (pp. 633, 634; 646, 647; 630, 643 and 636, respectively). A somewhat less transparent notation of energy and density fluctuations can be found in the form of the Fluctuation–Dissipation Theorem in the book [36] (p. 214). As can be seen, the thesis of the Old Fluctuation Theorem is missing the entropy fluctuation. This is probably due to the fear that the entropy fluctuation might violate the Second Law of Thermodynamics, so it is risky to talk about it. Nevertheless, we can heuristically assume that (with a three-quarter chance that the power does not get complicated here as for pressure)
(41)$$\frac{\langle S^{2}\rangle -{\langle S\rangle }^{2}}{{\langle S\rangle }^{2}}\overset{?}{\approx }\frac{1}{\langle N\rangle }.$$

This form of the square of the general relative fluctuation (but not exactly for the entropy) is given in the textbook by Landau and Lifschitz [62] (p. 24). This means that for systems with a small number of particles, entropy fluctuations can be relatively large, and this also applies to negative values. However, for large systems, the fluctuations, including negative ones, will be small but will not be equal to mathematical zero. Therefore, a categorical formulation of the Second Law of Thermodynamics in the framework of statistical mechanics is a difficult task (if at all feasible—in categorical terms; see [62] (p. 45).

## 2. Clarification of the Second Law of Thermodynamics Within Phenomenological Thermodynamics

### 2.1. Additional Definitions and Some Inequalities Between Entropy Types

The introduction to phenomenological thermodynamics begins with the definition of *Perpetuum Mobile* of Kind 0 and of the First and Second Kinds. It turns out that to better understand the Second Law of Thermodynamics, we need other kinds of *Perpetuum Mobile*.

**Definition 18** (*Perpetuum Mobile* of the Third Kind)**.**
*We call a hypothetical heat engine whose efficiency would exceed that of a reversible engine (i.e., the Carnot engine) a “Perpetuum Mobile” of the Third Kind. This hypothetical engine is the inverse of an irreversible refrigeration cycle which is possible to achieve in nature.*


**Definition 19** (*Perpetuum Mobile* of the Fourth Kind)**.**
*We call a hypothetical heat engine that extracts heat energy from the environment without significant temperature differences (in thermodynamic equilibrium) a “Perpetuum Mobile” of the Fourth Kind.*


**Definition 20** (Counterthermal *Perpetuum Mobile*)**.**
*A hypothetical heat engine in which the heat source (“heater”) has a lower temperature than the heat receiver (“cooler”) is called a Counterthermal “Perpetuum Mobile”.*


**Definition 21** (Refrigeration *Perpetuum Mobile*)**.**
*A hypothetical device operating in the refrigeration cycle (refrigerator and heat pump), whose efficiency is greater than the value allowed by the laws of thermodynamics, is a Refrigeration “Perpetuum Mobile”. It is generally the inverse of irreversible heat engines, i.e., engines with efficiency lower than the Carnot engine.*


Thanks to these definitions, it is possible to make some synthetic clarifications and achieve a deeper understanding of the Second Law of Thermodynamics. For example, the non-existence of a *Perpetuum Mobile* of the Third Kind (and not the other) fully expresses this principle [17]. However, the non-existence of a Refrigeration Perpetuum Mobile makes us realize something that most students and even teachers (including academics) did not know. In other words, inverse (refrigeration) cycles compared with engine cycles, e.g., Otto, Diesel, Stirling and Erriscon, do not physically exist (only on paper). As a result, this means greater restrictions on the coefficient of “efficiency” of heat pumps, which is still greater than 100% without violating the Second Law of Thermodynamics.

The non-existence of a *Perpetuum Mobile* of the Fourth Kind and the non-existence of a Counterthermal *Perpetuum Mobile* are more obvious. However, these concepts can be used to show the shortcomings of traditional statements. For example, the Kelvin–Ostwald Statement allows for a Counterthermal *Perpetuum Mobile* with an efficiency of less than 100% [17]. The *Perpetuum Mobile* of the Fourth Kind fills the gap in the definition of *Perpetuum Mobile* of the Second Kind. In other words, hypothetically, there could be an engine that absorbs heat from the surroundings that are in thermodynamic equilibrium, then converts part of it into work, and returns the rest of the heat to the surroundings.

Previously, three thermodynamic definitions of C-type, *Q*-type (or *q*-type) and *V*-type entropy were given. The C-type definition is a simplified definition adapted to two heat reservoirs. The definition of the *Q*-type is a typical integral definition. The entropies of the *V*- and *Q*-types are apparently equal, and the entropy of the C-type is clearly different, but under certain assumptions, some inequalities hold.

**Statement** **1** (*Q*- and C-Type Entropy Relation)**.**
*In the case of changes in a closed gas system (with a constant number of molecules) in thermal contact with two heat reservoirs, the outflow of the Q-type entropy from the gas is not greater than the C-type entropy change in the heat reservoirs, i.e.,*

(42)
$$-\Delta S_{Q}\leq \Delta S_{C},$$

*which indicates that the gas system can perform work and work can be performed on the system despite the thermal isolation of the entire reservoir system along with the working gas. Regardless of the work performed by the gas (or on the gas), we assume that at the elementary differential level, heat from the gas cannot flow to higher temperature or flow from lower temperature. This assumption is a reinforced version of Clausius’ First Statement in “local” form.*


**Proof.** For a gas temperature $T_{3}$ that is intermediate with respect to the heat reservoirs $T_{2}<T_{3}<T_{1}$, we can have standard heat flows $Q_{1}\geq 0$ and $Q_{2}\geq 0$. If the heat flowing into the gas δQ is divided into heat from the radiator ($\delta Q_{\left(1\right)}=\delta Q_{1}$) and from the cooler ($\delta Q_{\left(2\right)}=-\delta Q_{2}$), then we obtain the inequalities
(43)$$0\geq \int \frac{-\delta Q_{\left(1\right)}}{T_{3}}\leq \frac{-Q_{1}}{T_{1}}\leq 0,\;\;\;\;0\leq \int \frac{-\delta Q_{\left(2\right)}}{T_{3}}\leq \frac{Q_{2}}{T_{2}}\geq 0\;\;\;\;⟶\;\;\;\;-\Delta S_{Q}\leq \Delta S_{C}.$$ For a gas temperature $T_{3}$ higher than the heat reservoirs $T_{2}<T_{1}<T_{3}$, we can have heat flows to the reservoirs $Q_{1}\leq 0$ and $Q_{2}\geq 0$. This gives the inequalities
(44)$$0\leq \int \frac{-\delta Q_{\left(1\right)}}{T_{3}}\leq \frac{-Q_{1}}{T_{1}}\geq 0,\;\;\;\;0\leq \int \frac{-\delta Q_{\left(2\right)}}{T_{3}}\leq \frac{Q_{2}}{T_{2}}\geq 0\;\;\;\;⟶\;\;\;\;-\Delta S_{Q}\leq \Delta S_{C}.$$ For the gas temperature $T_{3}$ lower than the heat reservoirs $T_{3}<T_{2}<T_{1}$, we have heat outflows from the reservoirs $Q_{1}\geq 0$ and $Q_{2}\leq 0$. This gives the inequalities
(45)$$0\geq \int \frac{-\delta Q_{\left(1\right)}}{T_{3}}\leq \frac{-Q_{1}}{T_{1}}\leq 0,\;\;\;\;0\geq \int \frac{-\delta Q_{\left(2\right)}}{T_{3}}\leq \frac{Q_{2}}{T_{2}}\leq 0\;\;\;\;⟶\;\;\;\;-\Delta S_{Q}\leq \Delta S_{C}.$$ We also need to consider the temporary equality of the gas temperature with the temperature of some reservoir.For $T_{3}=T_{1}$, it is possible to remove heat from the gas and transfer it to the cooler $Q_{2}\geq 0$, but the sign of $Q_{1}$ can be any:
(46)$$\int \frac{-\delta Q_{\left(1\right)}}{T_{3}}=\frac{-Q_{1}}{T_{1}},\;\;\;\;0\leq \int \frac{-\delta Q_{\left(2\right)}}{T_{3}}\leq \frac{Q_{2}}{T_{2}}\geq 0\;\;\;\;⟶\;\;\;\;-\Delta S_{Q}\leq \Delta S_{C}.$$ For $T_{3}=T_{2}$ it is possible for the gas to absorb heat from the heater $Q_{1}\geq 0$, but the sign of $Q_{2}$ can be any:
(47)$$0\geq \int \frac{-\delta Q_{\left(1\right)}}{T_{3}}\leq \frac{-Q_{1}}{T_{1}}\leq 0,\;\;\;\;\int \frac{-\delta Q_{\left(2\right)}}{T_{3}}=\frac{Q_{2}}{T_{2}}\;\;\;\;⟶\;\;\;\;-\Delta S_{Q}\leq \Delta S_{C}.$$ Finally, heat flow bypassing the gas is possible (δQ=0), in which $Q_{1}=Q_{2}\geq 0$:
(48)$$0=\int \frac{-\delta Q}{T_{3}}\leq \frac{-Q_{1}}{T_{1}}+\frac{Q_{2}}{T_{2}}\geq 0\;\;\;\;⟶\;\;\;\;-\Delta S_{Q}\leq \Delta S_{C}.$$ The performance of positive or negative work by the gas does not directly affect the heat flows but may change the temperature $T_{3}$ of the gas (in isothermal compression or expansion). Therefore, it can only change in one of the cases considered above.The last case is the conversion of work into heat supplied to the reservoirs $Q_{1}=-W_{1}\leq 0$ and $Q_{2}=W_{2}\geq 0$ without the use of gas:
(49)$$0=\int \frac{-\delta Q}{T_{3}}\leq \frac{W_{1}}{T_{1}}+\frac{W_{2}}{T_{2}}\geq 0\;\;\;\;⟶\;\;\;\;-\Delta S_{Q}\leq \Delta S_{C}.$$ This gas-mediated effect should also be considered. The work is then used for adiabatic compression of the gas, which leads to an increase in its temperature $T_{3}$. Then, we release this work transferred to the gas in the form of heat according to one of the already indicated cases in which the proved inequality is satisfied. □

Despite the multitude of cases considered in the proof, two conclusions follow from the above Statement 1. The first one is a simple, synthetic request, while the second one is more serious and uses Statement 2 below.

**Corollary 1** (Out–Inflow Entropy Relation)**.**
*During the natural outflow of heat δQ>0 from a body with a temperature $T_{out}$ not lower than the temperature $T_{int}$ of the body receiving heat ($T_{out}\geq T_{int}$), the outflow of entropy is not greater than the inflow of entropy:*

(50)
$$-dS_{out}\leq dS_{int}\;\;\;\;⟷\;\;\;\;-\frac{-\delta Q}{T_{out}}\leq \frac{\delta Q}{T_{int}},$$

*where equality can only occur at equal temperatures, i.e., in an isothermal process.*


**Proof.** Proving a rough inequality is trivial because both sides of the inequality are positive, and the denominator of the right side is no larger than that of the left side, so the fraction of the right side is no smaller:
(51)$$0<\frac{\delta Q}{T_{out}}\leq \frac{\delta Q}{T_{int}}>0\;\;\;\;\mathrm{for}\;\;\;\;T_{out}\geq T_{int}.$$ However, for $T_{out}=T_{int}$, of course, the outflow and inflow of entropy are equal. □

Note that Corollary 1 is differential in nature, so its thesis will also be valid for overall entropy changes. At the same time, this corollary is synonymous with the inequality for the entropy of the *Q*- and *q*-types given in Definition 7. Corollary 1 and its broader version Statement 1, as well as Statement 2 below, raise an important conclusion reminiscent of the Second Law of Thermodynamics.

**Corollary 2** (Deduction of the Second Law of Thermodynamics)**.**
*If the working gas in thermal contact with two heat reservoirs operates in cyclic mode and a “local” enhanced Clausius’ First Statement occurs, the C-type entropy of the heat reservoirs cannot decrease:*



(52)
$$\Delta S_{C}\geq 0.$$


**Proof.** In cyclic processes, the *V*-type entropy change in the gas is zero. This is due to the fact that such entropy is an integral of the complete differential (see further Statement 2):
(53)$$\Delta S_{V}=\oint \frac{dE+pdV}{T}=\oint dS_{V}=S_{V}-S_{V}=0.$$It is reasonable to assume that the working gas is not subject to vacuum expansion or other versions of irreversible adiabatic transformation (see Statement 2), so in a cyclic process,
(54)$$\Delta S_{Q}=\Delta S_{V}=0.$$By assuming a “local” enhanced version of Clausius’ First Statement, we can use the thesis of Statement 1 and obtain the proven thesis:
(55)$$-\Delta S_{Q}\leq \Delta S_{C}\;\;\;\;\overset{\Delta S_{Q}=0}{⟶}\;\;\;\;0\leq \Delta S_{C}.$$□

Note that even without demonstrating the zeroing of $\Delta S_{Q}$, Statement 1 could be expressed in a form resembling the Second Law of Thermodynamics for the system:(56)$$\Delta S_{Q}+\Delta S_{C}\geq 0.$$

So, how is it possible that we so easily obtain one or another relation of the Second Law of Thermodynamics? What assumption led to these results? The main assumption was that the “local” (at the differential level) heat flow to or from the gas cannot take place from lower to higher temperature.

So, how does this assumption differ from Clausius’ First Statement? It seems that the difference is very subtle. In other words, applying Clausius’ First Statement globally to processes taking place between heat reservoirs does not result in the exhaustive Second Law of Thermodynamics in the form $\Delta S_{C}\geq 0$ [17]. However, by applying the “locally” enhanced Clausius’ First Statement to the differential heat flows of an ideal gas, it is possible to deduce the $\Delta S_{C}\geq 0$ principle as a simple conclusion from Statement 1.

Therefore, the “local” understanding based on a gaseous working medium with variable temperature differs from the “global” understanding based only on the effective operation of heat engines and refrigerators. These results are somewhat surprising and require even more detailed interpretations and analyses. Undoubtedly, the “local” result resembles the so-called Clausius’ inequality $\Delta S_{q}\leq 0$ (see descriptions in Definitions 7 and 9) with a slightly different heat sign convention and a different temperature reference. However, given the equation $\Delta S_{V}=0$ (or $\Delta S_{Q}=0$) for the cyclic gas process and the introduction of entropy $\Delta S_{C}\geq 0$ for reservoirs, the considered “local” result may appear as a refinement of the Clausius’ inequality issue. The result below is of a slightly different nature, which is why it is included at the end of this section.

**Statement** **2** (*Q*-Type and V-Type Entropy Relation)**.**
*In the case of changes in a closed gas system (with a constant number of molecules), the entropy change calculated according to the Q-type and V-type entropy formulas satisfies the inequality*

(57)
$$\Delta S_{Q}\leq \Delta S_{V}\;\;\;\;⟷\;\;\;\;\int \frac{\delta Q}{T}\leq \int \frac{dE+pdV}{T},$$

*which is a sharp inequality for the irreversible adiabatic process (*

$\Delta S_{Q}<\Delta S_{V}$

*) and in particular for the process of expansion into vacuum. The equality holds, of course, at least for reversible processes.*


**Proof.** According to the First Law of Thermodynamics, which is the Law of Conservation of Energy or even the definition of heat, we have
(58)δQ=dE−δW,
where neither heat nor work are exact differentials. The mechanical definition of work leads to the following expression for the work of a gas (δW means work on the gas):
(59)−δW:=pdV,
which refers to a quasi-static process. If the process takes place very quickly, the gas molecules will not have time to perform negative work—that is, they will not have time to lose energy or part of it. However, the opposite process of spontaneous gas collapse is not possible, so it does not need to be considered. So, we are left with the inequality
(60)−δW≤pdV,
which will be a sharp inequality for every version of the irreversible adiabatic process (−δW<pdV). In the extreme case of expansion into a vacuum, the left side is zero, and the right side is positive. Although the right side of the inequality is not always work (for which the equality of the first law is always satisfied), the right side replacing work in the *V*-type entropy formula gives a more general entropy formula than the *Q*-type entropy. This is due to the fact that the entropy differential
(61)$$dS_{V}=\frac{dE+pdV}{T},$$
is a complete differential (we assume N=const) thanks to the integrating factor 1/T. To verify this, we need the following equations: Clapeyron $pV=Nk_{B}T$ and energy $E=3Nk_{B}T/2$. Thanks to them, it is possible to integrate the entropy for a “monatomic” ideal gas, i.e.,
(62)$$S_{V}=\frac{3}{2}Nk_{B}\ln E+Nk_{B}\ln V+\mathrm{const},$$
which is basically consistent with the $S_{\Gamma }$ entropy formula, omitting the constants that are irrelevant here (including the terms with just *N*). Surprisingly, this is the formula for the entropy of a gas in a state of thermodynamic equilibrium, despite previous considerations of irreversible adiabatic processes or expansion into a vacuum. Simply, the formula $\Delta S_{V}$ with complete differential covers irreversible adiabatic processes, and the formula $\Delta S_{Q}$ with heat does not describe these cases. □

The occurrence of a weak or sharp inequality instead of an equality does not mean a violation of the standard ordinary First Law of Thermodynamics. This only means that the expression for the work δW performed by the working factor may be smaller than its usual formula δW≤pdV and occurs, for example, in the case of expansion into a vacuum, where work is zero. Such a refinement of Clausius’ entropy definition formula was provided by one of the authors of this work in 2018 in the publication [34], and it is given in the entry “Entropy” of the online version of the Britannica encyclopedia (https://www.britannica.com/science/entropy-physics (accessed on 19 December 2024)).

### 2.2. Carnot Type Statement Instead of Kelvin–Ostwald Statement

Ostwald formulated the Kelvin Statement in the language *Perpetuum Mobile* of the Second Kind (see Definition 2). That is the reason why such a formulation is called the Kelvin–Ostwald Statement here (see Definition 5). However, for an exhaustive formulation of the Second Law of Thermodynamics, we need the *Perpetuum Mobile* of the Third Kind, not the *Perpetuum Mobile* of the Second Kind. This misunderstanding has already grown deep into the fabric of the status quo of physical knowledge, and it is very difficult to eradicate this erroneous paradigm. In any case, this issue was thoroughly researched and proven by one co-author of this work in the publication [17]. In this respect, note that in the recent Ref. [14], Ostwald’s formulation, together with the weakness and the not full equivalence of the Kelvin–Ostwald Statement (called the Kelvin–Planck Statement) with Clausius’ First Statement and with the Second Law of Thermodynamics, was not discussed.

Even the most skeptical first reviewer of this publication (the reviews are available together with the publication on the journal’s website) claims that all specialists know that the Kelvin Statement (i.e., the Kelvin–Ostwald Statement) is not fully equivalent to the Second Law of Thermodynamics. However, it must be stated that this is not reflected in the state of the literature on the subject, so the problem should not be underestimated.

It is worth pointing out the reason why the Kelvin–Ostwald Statement (Kelvin Statement) is not exhaustive, although it is true. In other words, in the publication [17], it was shown that the logical structure of the Kelvin–Ostwald Statement allows for five types of models. Even the best of these models, denoted by ${\left\{K\right\}}_{0<\eta_{m}<1}^{↓}$, containing the most efficient reversible engine, does not in any way determine the value of the maximum efficiency (Carnot efficiency). In addition, the statement under consideration allows for very non-physical and unattractive models. An example is a model denoted by ${\left\{K\right\}}_{0}^{↕}$, in which heat flows in either direction and the (positive) work is not performed by the system—although any (negative) work can be performed over the system. This model includes the Refrigeration *Perpetuum Mobile*, if we consider this device strange at all in a world where heat flows in any direction. Another model, denoted by ${\left\{K\right\}}_{1^{-}}^{↓}$, is completely devoid of cooling processes, and within it, there is no maximum efficiency of engines, because any efficiency lower than 1 (100%) is permissible. The other two models, ${\left\{K\right\}}_{1^{-}}^{↑}$ and ${\left\{K\right\}}_{0<\eta_{m}<1}^{↑}$, include the Counterthermal *Perpetuum Mobile*.

In this situation, the Kelvin–Ostwald Statement should be strengthened by replacing the *Perpetuum Mobile* of the Second Kind with the *Perpetuum Mobile* of the Third Kind in its content. This has already been partially performed in Ref. [17], but it is useful to propose it explicitly.

The proposal to formulate a principle of physics is not subject to proof, just as physical principles are not subject to mathematical proof. Nevertheless, it is possible to formulate a statement about the equivalence of a given formulation of a physical principle with another formulation and provide evidence of such a statement:

**Proposition 1** (Carnot Type Statement)**.**
*There is no “Perpetuum Mobile” of the Third Kind. In other words, each cyclically operating heat engine ($W>0,\;Q_{1}>0$) has efficiency not greater than that of a reversible engine (Carnot engine), i.e.,*

(63)
$$\frac{W}{Q_{1}}=:\eta \leq \eta_{C}:=\frac{T_{1}-T_{2}}{T_{1}}.$$

*All thermodynamic processes are allowed, the combination of which (addition) does not lead to the formation of a “Perpetuum Mobile” of the Third Kind (violation of inequality (Equation 63) with the assumptions for the engine $W>0,\;Q_{1}>0$). Individual processes that are components of a sum of processes must satisfy the First Law of Thermodynamics, but they do not have to satisfy the assumption for engines (they do not have to be engines).*


**Statement** **3** (Equivalence of Carnot Type Statement with Clausius’ Second Statement)**.**
*The Carnot Type Statement (Proposition 1) with all its conditions is equivalent to the Second Law of Thermodynamics in the formulation of the law of entropy increase, i.e., Clausius’ Second Statement (Definition 9).*


**Proof.** First, we show that the law of increasing entropy (Equation 6) gives rise to the Carnot Type Statement (Equation 63). For this purpose, we transform inequality (Equation 6) into inequality (Equation 63) by respecting its assumption ($Q_{1}>0$):
(64)$$\frac{-Q_{1}}{T_{1}}+\frac{Q_{2}}{T_{2}}\geq 0\;\;\;\overset{Q_{1}>0}{\underset{T_{2}>0}{\to }}\;\;\;\frac{Q_{2}}{Q_{1}}\geq \frac{T_{2}}{T_{1}}\;\;\;⟶\;\;\;1-\frac{Q_{2}}{Q_{1}}=\eta \leq 1-\frac{T_{2}}{T_{1}}=\eta_{C}.$$Now, we need to demonstrate a more difficult implication that goes the other way. Let us assume that the process $\left(Q_{1}>0,W,Q_{2}\right)$ is a Carnot engine with efficiency $\eta_{C}$. Consider more of an arbitrary process $\left(q_{1},w,q_{2}\right)$ that respects a priori only the First Law of Thermodynamics of the form $q_{1}=w+q_{2}$. We now want to combine the considered processes, but so that $Q_{1}+q_{1}>0$. A single connection to the Carnot engine does not guarantee that this inequality is met. However, there is always a natural number *k* such that $kQ_{1}+q_{1}>0$. Therefore, for the composition of *k* Carnot engines with the process $\left(q_{1},w,q_{2}\right)$, the Carnot condition should apply:
(65)$$1-\frac{kQ_{2}+q_{2}}{kQ_{1}+q_{1}}\leq 1-\frac{T_{2}}{T_{1}}\;\;\;\overset{kQ_{1}+q_{1}>0}{\to }\;\;\;kQ_{2}+q_{2}\geq \frac{T_{2}}{T_{1}}\left(kQ_{1}+q_{1}\right).$$ The terms containing *k* in the last inequality are for the Carnot engine, so they are equal and cancel out in this inequality. From the remaining part, we obtain the condition for the change in entropy of the additional process:
(66)$$q_{2}\geq \frac{T_{2}}{T_{1}}q_{1}\;\;\;\overset{T_{2}>0}{\to }\;\;\;\frac{q_{2}}{T_{2}}-\frac{q_{1}}{T_{1}}\geq 0.$$ This completes the proof of the second implication, and thus ends the proof of the equivalence theorem. □

In addition to the basically non-existent *Perpetuum Mobile* of the Third Kind, a key role in Proposition 1 is played by the Carnot cycle, or equivalently, the Carnot engine. The question then becomes the following: is it possible to build a real heat engine whose reference cycle would essentially be the Carnot cycle (not the completely perfect Carnot cycle with sharp peaks)?

There are no theoretical reasons why such a Carnot engine cannot be made. Moreover, one should basically know how such an engine could be built technically. This requires some specific solutions that are not available, for example, in Stirling and Ericsson engines, because they do not operate in Carnot cycle mode. This solution does not rely on a heat refrigerator in the Stirling engine, which cannot be effective due to the Second Law of Thermodynamics—heat cannot be arbitrarily recycled and stored despite existing temperature differences.

There are at least two known patents for the Carnot engine: a US patent from 1991 [63] and a patent filed in 2013 at the Chinese patent office [64]. However, according to the authors of this work, neither of these patents describes an engine that would be able to operate in the comparative Carnot cycle.

Moreover, there is an opinion in the scientific community that the Carnot engine is only a theoretical model of an ideal engine and it can only be used to compare efficiency with real engines. This belief, however, does not result from any principle of thermodynamics and, in a sense, introduces an element of fiction and absurdity into science. In other words, if the Carnot engine would fundamentally not exist, why is the science of thermodynamics and the science of heat engines based largely on the Carnot cycle or engine? It is worth remembering this in the context of 2024, which is the 200th anniversary of Carnot’s only publication in 1824 [65].

### 2.3. The Inequality of Heat and Temperature Proportions Instead of Clausius’ First Statement

It turns out that Clausius’s first attempt to formulate the Second Law of Thermodynamics was not exhaustive. Clausius’ First Statement (Definition 4) is correct and true, but it does not fully reflect the principle of entropy increase. In other words, Clausius’ First Statement is a semi-quantitative (almost qualitative) principle about the direction of thermodynamic phenomena, but it is not an exhaustive quantitative principle. It is worth noting that we are now talking about the principle understood “globally” in relation to heat reservoirs and not in the “local” understanding applied to working gas (see Section 2.1 above and further references below).

The formal logical status of Clausius’ First Statement structure was explicitly presented and proven by one of the authors of this work in the base publication [17]. It was proven to be not only non-equivalent to the entropy increase principle but also strictly non-equivalent to the Kelvin–Ostwald Statement (Kelvin Statement).

It turned out that Clausius’ First Statement allows for a class of models, marked ${\left\{CI\right\}}_{\eta_{m}}$, in which the maximum efficiency takes any value in the range $0\leq \eta_{m}\leq 1$ [17]. Therefore, in this class of models, there is a model containing the *Perpetuum Mobile* of the Second Kind ($\eta_{m}=1$), in which no refrigeration processes occur, so it does not violate Clausius’ First Statement. In such a ${\left\{CI\right\}}_{1}$ model, the only reversible process is the *Perpetuum Mobile* of the Second Kind with 100% efficiency. The inverse process here is the conversion of work into heat of the radiator, so this inverse process cannot be called a refrigeration process due to $Q_{2}=0$ instead of $Q_{2}<0$, corresponding to the refrigerator. The remaining models ($\eta_{m}<1$) consistent with Clausius’ First Statement are consistent with the Kelvin–Ostwald Statement. Such models feature a reversible engine with a maximum efficiency of less than 100%. However, this maximum efficiency is not further limited or defined in any way. Therefore, on one hand, such an engine could be a *Perpetuum Mobile* of the Third Kind, and on the other hand, its efficiency (in a specific model) could be accidentally too low compared with the Carnot cycle. In both cases, we cannot speak of equivalence with the full Second Law of Thermodynamics. Moreover, Clausius’ First Statement allows for a model at the other extreme, i.e., the ${\left\{CI\right\}}_{0}$ model. This model does not contain any heat engines, but it does contain ideal refrigerators, i.e., the Refrigeration *Perpetuum Mobile*.

Since Clausius’ First Statement with a “global” reference to two heat reservoirs turned out to be a non-exhaustive formulation, is it possible to somehow quantitatively specify this principle of spontaneous heat flow towards the lower temperature of the cooler? It turns out that in a sense it is possible. For this purpose, it is enough to introduce an inequality in the heat–temperature ratio, which will force a sufficiently large heat transfer from the radiator in relation to the heat discharged by the radiator. This quantitative reinforcement of qualitative Clausius’ First Statement will prevent the occurrence of the *Perpetuum Mobile* of the Third Kind, in which too little heat flows to the radiator. Here is the phrase we are looking for.

**Proposition 2** (Inequality of Heat and Temperature Proportions). *If the working medium is in thermal contact only with two heat reservoirs with fixed absolute temperatures $T_{1}>T_{2}$ and $T_{2}>0$, then the process of extracting heat $Q_{1}>0$ from the radiator must satisfy the Inequality of Heat and Temperature Proportions:*
(67)$$\frac{Q_{2}}{Q_{1}}\geq \frac{T_{2}}{T_{1}}\;\;\;\;for\;\;\;\;Q_{1}>0,$$
*where $Q_{2}$ is the heat absorbed by the cooler in this process. The above inequality defines a set of base processes (mainly engine, but not exclusively). Moreover, any thermal process will be possible, with parameters marked with lowercase letters, which, when added to each base process, will also be a base process while maintaining the condition of positive denominator:*
(68)$$\frac{Q_{2}^{\prime }}{Q_{1}^{\prime }}=\frac{Q_{2}+q_{2}}{Q_{1}+q_{1}}\geq \frac{T_{2}}{T_{1}}\;\;\;\;for\;\;\;\;Q_{1}^{\prime }=Q_{1}+q_{1}>0.$$
*In other words, the Inequality of Heat and Temperature Proportions applied to appropriately added processes determines all possible thermal processes. However, mechanical work results here from the First Law of Thermodynamics ($w=q_{1}-q_{2}$) and is not subject to any additional conditions.*

It can be seen that mathematically, the above inequality is close to the inequality of the Carnot Type Statement (Proposition 1). However, this interpretation is closer to Clausius’ First Statement. Here, we have an inequality of the heat ratio resulting from the temperature ratio (less than unity). The inequality sign means that the heat flow to the cooler is basically unlimited from above in the basic processes themselves (at the expense of any negative work $Q_{2}=Q_{1}-W$). However, the heat $Q_{2}$ is primarily limited from the bottom—an appropriate amount of it simply must flow to the cooler in the base process.

It is worth noting the specific simplicity of the structure of the conditions in Proposition 2. In practice, they do not include the concept of work or additional concepts such as efficiency or entropy. In Proposition 2, only heat and temperature terms appear, perhaps in the simplest possible configuration in the form of ratios of quantities of the same kind. This inequality of proportions is reminiscent of a beautiful and little-known ancient mathematical construction, given below.

Eudoxus of Cnidus in the 4th century BC managed to practically define all positive real numbers by using appropriate inequalities and equalities regarding the appropriate ratios of natural numbers [66]. The equalities corresponded to rational numbers, and sharp inequalities described irrational numbers. However, in Proposition 2, the equality describes a reversible process (Carnot cycle), and the sharp inequalities describe irreversible processes. The cited inspiration taken from Eudoxus of Cnidus (and Mark Kordos—the author of the article [66]) makes us aware that the proof of the equivalence of the Inequality of Heat and Temperature Proportions requires the use of additional formal conditions contained in Proposition 2.

**Statement** **4** (Equivalence of Inequality of Heat and Temperature Proportions with the Second Law of Thermodynamics)**.**
*The Inequality of Heat and Temperature Proportions (Proposition 2) with all its conditions is equivalent to the Second Law of Thermodynamics in the Carnot Type Statement on the non-existence of the “Perpetuum Mobile” of the Third Kind (Proposition 1).*


**Proof.** In the proof of the equivalence of Proposition 2 with Proposition 1, we will use Statement 3, according to which Proposition 1 is equivalent to the law of non-decreasing entropy, i.e., Definition 9 (Clausius’ Second Statement). Therefore, we prove the equivalence of Proposition 2 with Definition 9. First, we prove that Proposition 2 follows from Definition 9 (with the condition $Q_{1}>0$ from Proposition 2):
(69)$$\frac{-Q_{1}}{T_{1}}+\frac{Q_{2}}{T_{2}}\geq 0\;\;\;\;\overset{Q_{1}>0}{\underset{T_{2}>0}{\to }}\;\;\;\;\frac{Q_{2}}{Q_{1}}\geq \frac{T_{2}}{T_{1}}.$$Now, we prove that Definition 9 (Clausius’ Second Statement) follows from Proposition 2. We start with the simplest case of base processes ($Q_{1}>0$):
(70)$$\frac{Q_{2}}{Q_{1}}\geq \frac{T_{2}}{T_{1}}\;\;\;\;\overset{Q_{1}>0}{\underset{T_{2}>0}{\to }}\;\;\;\;\frac{-Q_{1}}{T_{1}}+\frac{Q_{2}}{T_{2}}\geq 0.$$ Now, let us consider basically any process added to the base process (the condition $Q_{1}+q_{1}>0$ can always be met by selecting a base process with a large $Q_{1}$ or by multiplying any base process):
(71)$$\frac{Q_{2}+q_{2}}{Q_{1}+q_{1}}\geq \frac{T_{2}}{T_{1}}\;\;\;\;\overset{Q_{1}+q_{1}>0}{\underset{T_{2}>0}{\to }}\;\;\;\;\frac{-Q_{1}-q_{1}}{T_{1}}+\frac{Q_{2}+q_{2}}{T_{2}}\geq 0.$$ If we choose the Carnot cycle as the basic process, i.e., a reversible process for which equality holds (we can choose the Carnot cycle, because the examined inequality is to be true for all basic processes for which $Q_{1}+q_{1}>0$),
(72)$$\frac{Q_{2}}{Q_{1}}=\frac{T_{2}}{T_{1}},$$
then we obtain the desired thesis for any process:
(73)$$\frac{-q_{1}}{T_{1}}+\frac{q_{2}}{T_{2}}\geq 0.$$Applying the condition (Equation 67) to base processes other than the Carnot cycle obviously leads to weaker but consistent inequalities. Let us write it schematically and briefly, still referring to the designation of the Carnot cycle:
(74)$$\frac{Q_{2}+\delta }{Q_{1}}>\frac{T_{2}}{T_{1}}\;\;\;\;\ldots ⟶\;\;\;\;\frac{-q_{1}}{T_{1}}+\frac{q_{2}+\delta }{T_{2}}\geq 0,$$
(75)$$\frac{Q_{2}}{Q_{1}-\delta }>\frac{T_{2}}{T_{1}}\;\;\;\;\ldots ⟶\;\;\;\;\frac{-q_{1}+\delta }{T_{1}}+\frac{q_{2}}{T_{2}}\geq 0,$$ where δ>0 and, in the second case, it is assumed that $Q_{1}-\delta >0$ and $Q_{1}-\delta +q_{1}>0$. Indeed, these two versions are weaker inequalities, so including all inequalities leads to the correct inequality for the C-type entropy. □

Now that Clausius’ First Statement has been clarified, it is worth answering the following question: has anything changed in terms of the direction of heat flow? Qualitatively not—heat still flows spontaneously towards the lower temperature. However, quantitatively, the heat flow towards the lower temperature must be sufficiently large (taking into account that the rest of the heat can be converted into work). This value is determined by Inequality of Heat and Temperature Proportions (Equation 67). But how can we explain the operation of refrigerators or heat pumps, where heat flow is forced against the temperature difference? It can also be explained by the Inequality of Heat and Temperature Proportions but used to connect the base process with the tested process (Equation 68). Then, all three values $q_{1}$, $q_{2}$ and $w=q_{1}-q_{2}$ can be negative without breaking inequalities (Equation 68).

In fact, even in refrigeration devices, heat flows elementally (“locally”) from one temperature to another through the working gases. The point is that as a result of adiabatic processes, the working medium is given an appropriate temperature—a lower temperature to absorb heat and a higher temperature to release heat. By using the working medium prepared in this way, a refrigerator or heat pump can carry out processes seemingly (“globally”) contrary to Clausius’ First Statement.

This issue emerged in Section 2.1 in the form of a “local” enhancement of Clausius’ First Statement (without explicit definition), which would aspire to equivalence with the Second Law of Thermodynamics. However, there are doubts whether the temperature inequality relation itself does not have additive freedom, which will distort the entropy value and the Carnot efficiency:(76)$$T_{1}\geq T_{2}\;\;\;\;⟶\;\;\;\;T_{1}+T_{0}\geq T_{2}+T_{0}\;\;\;\;⟶\;\;\;\;\Delta S_{C}^{\prime }=\frac{-Q_{1}}{T_{1}+T_{0}}+\frac{Q_{2}}{T_{1}+T_{0}},$$
(77)$$T_{1}\geq T_{2}\;\;\;\;⟶\;\;\;\;T_{1}+T_{0}\geq T_{2}+T_{0}\;\;\;\;⟶\;\;\;\;\eta_{C}^{\prime }=\frac{T_{1}-T_{2}}{T_{1}+T_{0}}.$$

In other words, any direct Clausius’ First Statement amplification (even “local”) seems to be immune to the above freedom to “choke” the absolute temperature. However, this does not apply to the Inequality of Heat and Temperature Proportions.

### 2.4. Negative or Infinitesimal Hyperreal Absolute Temperature?

It is worth mentioning a very unusual (purely formal) exception to the principle of heat flow towards a lower temperature and a proposal to remedy it. Readers who do not like examples of abstract mathematics in physics can skip this section without major detriment to the understanding of this article as a whole.

According to the thermodynamic definition of temperature, defined by the inverse of the derivative of entropy with respect to energy (and not the average value of the particle’s energy), it is possible to have a negative “absolute” temperature on the Kelvin scale. The issue is known from the turn of the 1940s and 1950s [67], and in the last decade, even experiments with quantum states at negative temperatures were reported [68]. This situation occurs when the entropy of the system decreases as the energy of the laser-trapped states increases. However, such states tend to get rid of “excess” energy (heat) rather than absorb it. Informally, this means that such a negative “absolute” temperature is greater than any arbitrarily large positive temperature—in a sense even greater than plus infinity. Formally, it looks as if heat flows from a lower negative temperature to a higher one.

We can try to illustrate this exception with a partially formalized mathematical picture. Let us consider a convergent geometric series symbolizing temperature, which will be parameterized by *x* (for simplicity, we omit physical units):(78)$$T\left(x\right)=1+x+x^{2}+x^{3}+...=\frac{1}{1-x}\;\;\;\;\left(\mathrm{for}\;\;\;-1<x<1\right).$$

By increasing the *x* parameter, we increase the value of the conventional temperature T(x). In this way, we can obtain any high temperature. However, by decreasing *x*, we can reduce the temperature to 1/2. If we want to fill the missing temperature interval 0>T≥1/2, we must extend *x* to all negative values:(79)$$T\left(x\right)=......=\frac{1}{1-x}\;\;\;\;\left(\mathrm{for}\;\;\;-\infty <x<1\right).$$

We still have a well-defined function *x*, but the series for the added interval diverges. However, it is an alternating series, so assigning it a finite value does not contradict common sense. The most important thing is that the majority relationship in the set of *x* values carries over to the set of *T* values.

We can, therefore, transfer this hierarchy of the temperature axis order to the compatible order of the parameter *x*:(80)$$T\left(x_{1}\right)≺T\left(x_{2}\right)\;\;\equiv \;\;x_{1}\;<\;x_{2}.$$

So far, such a transfer does not give us anything new. However, let us extend the range of the parameter *x* to all possible real values:(81)$$T\left(x\right)=......=\frac{1}{1-x}\;\;\;\;\;\left(\mathrm{for}\;\;\;x\neq 1\right).$$

This procedure introduces negative temperatures for x>1. Note that negative temperatures appear for larger values of *x* than positive temperatures. Therefore, according to the new order relation “≺”, these “supposedly” negative temperatures are in fact physically greater than all positive ones.

But what role does the divergent series play in all this? Its role is crucial in this section. First of all, for x>1, the series, “despite the discrepancies”, seems to be a very large positive number, so it is not surprising that the value assigned to it (formally negative) is treated as “physically” greater than other positive values. Secondly, the series for x>1 diverges to +∞, so this constitutes the use of hyperreal infinity (see below). It can be said that due to the lack of hyperreal numbers in standard analysis, ordinary mathematics tries to use the free and in this case negative numbers to assign values to divergent series. Such extensions were analyzed by, for example, Leonard Euler and Srinivasa Ramanujan.

Before considering the next step, let us first take a look at the physically correct hierarchy of the order on the extended absolute temperature axis (which is not an ordinary mathematical order) with respect to the ordinary order on the *x*-axis:(82)$$T:\;\;\;\;0^{+}\;≺\;\;\frac{1}{2}\;≺\;1\;≺\;2\;≺\;+\infty \;≺\;-\infty \;≺\;-2\;≺\;-1\;≺\;-\frac{1}{2}\;≺\;\;0^{-},$$
(83)$$x:\;\;-\infty \;<\;-1\;<\;0\;<\;\frac{1}{2}\;<\;\;1^{-}\;\;<\;\;1^{+}\;\;<\;\;\;\frac{3}{2}\;\;<\;\;\;2\;\;<\;\;3\;\;<+\infty .$$

Note that the key parameter *x* is expressed as follows in terms of temperature:(84)$$x=1-\frac{1}{T},$$
so in the context of the order relation on the extended absolute temperature axis, the one could be omitted, yielding
(85)$$\tau =-\frac{1}{T}.$$ Indeed, such a parameter is often used for negative absolute temperatures [69], which has some justification in the thermodynamic definition of temperature (this definition directly determines the inverse of temperature). However, for power series analyses, *x* is more convenient than τ.

The corrected order hierarchy of the extended temperature axis differs from the real axis in the displacement of the negative ray after the positive ray. This procedure can be formalized even further by redefining negative temperatures into infinitesimally large positive numbers belonging to the so-called galaxy of hyperreal numbers (strictly speaking the left half of this galaxy):(86)$$T<0\;⟶\;T^{*}=T+\omega^{*}\;\;\;\mathrm{where}:\;\;\;T^{*},\omega^{*}\in \;R^{*}\;\;\;\mathrm{and}\;\;\;\left[\omega^{*}\right]=+\infty ,$$

However, one should always use one specific infinite hyperreal number $\omega^{*}$, e.g., represented by the simplest sequence diverging to infinity $\omega^{*}=\left[\left(a_{n}\right)\right]=\left[\left(n\right)\right]$. This understanding of the extended absolute temperature axis contains a more natural order (than the negative temperature axis) and, at the same time, formally eliminates negative temperatures from the theory and eliminates the exception of heat flow to higher temperatures.

Reinterpreting negative temperatures in terms of different infinitesimally large temperatures allows us to preserve the meaning of the Carnot engine efficiency formula and avoid the *Perpetuum Mobile* of the First Kind in the context of these negative temperatures. In other words, for a radiator with a negative temperature $T_{1}<0$, the engine would formally have an efficiency greater than 100%:(87)$$\eta =1-\frac{T_{2}}{T_{1}}>100\%\;\;\;\mathrm{for}\;\;\;T_{1}<0,$$

The radiator should have a negative temperature assigned here, because it is from where heat flows. Moreover, we would obtain the mathematical effect of efficiency greater than one if the cooler had a negative Kelvin temperature—but that would no longer make any physical sense.

If, instead of a negative number, we use an infinitely large number for the radiator, the efficiency will be practically equal to 100% (in the real domain and in the hyperreal domain, it will be infinitesimal smaller):(88)$$\eta =1-\frac{T_{2}}{T_{1}^{*}}\equiv 100\%-\varepsilon^{*}=100\%\;\;\;\mathrm{for}\;\;\;\left[T_{1}^{*}\right]=+\infty ,\;\;\left[\varepsilon^{*}\right]=0.$$

So, we have reached a strange situation, but not logically contradictory, in which the so-called Carnot engine (with negative/infinite hyperreal radiator temperature) is a *Perpetuum Mobile* of the Second Kind, and the definition of *Perpetuum Mobile* of the Third Kind here practically means the same as the *Perpetuum Mobile* of the First Kind.

However, strictly speaking (in the hyperreal domain), the order of the Carnot engine and the three mentioned *Perpetuum Mobile* machines by efficiency is preserved. The least efficient of them is the Carnot engine, then the *Perpetuum Mobile* of the Third Kind and then the *Perpetuum Mobile* of the Second Kind, and the highest efficiency is that of a completely aphysical *Perpetuum Mobile* of the First Kind. Excluding the last machine, the efficiency of the first three, in this unusual case (for negative/infinite hyperreal radiator temperature), differs infinitesimally little.

It is worth realizing at this point that Kelvin developed his absolute temperature scale based on the analysis of the efficiency function of the Carnot engine.

## 3. Clarification of the Second Law of Thermodynamics in Statistical Physics

### 3.1. Mean Value Study for Gibbs and Boltzmann Entropies

Note that the Gibbs (or Gibbs–Shannon) entropy is, by definition, the mean value minus the logarithm of the single probability:(89)$$S_{G}=-k_{B}<\ln p>.$$

This raises the question whether entropy does not have only an average statistical character. Of course, this could be said about almost all quantities of statistical physics, but there may be something more at stake here. Let us, therefore, consider the elementary entropy that we can assign to a specific state or a certain set of states described by probability $p_{i}$:(90)$$S_{i}=-k_{B}\ln p_{i}.$$

The ability to determine such entropy for a system would require knowledge that this system is in the “*i*” state. The probability of this state is $p_{i}$, but the assumption that the system is already in this state leads to the conditional probability p(i|i)=1. This is a bit like the problem of wavefunction collapse in quantum mechanics. Perhaps the problem of mixed states in statistical mechanics is not that different from quantum superposition. In any case, the system being in the “*i*” state does not depreciate the value of the probability $p_{i}$. Just as rolling a “5” on a six-sided dice does not depreciate the fact that the event had (has) a probability of one-sixth. Therefore, it seems that considering the $S_{i}$ entropy at a more fundamental level of the “pure *i*” state and not for the set of “mixed” states ${\left\{i\right\}}_{i}$ makes sense. Then, less detailed knowledge about the state of the system is the expected value from the available entropy value possibilities:(91)$$S_{G}=\langle S\rangle =\sum_{i}p_{i}S_{i}.$$

In a situation where “pure” states have equal probabilities (and therefore equal entropies), the average entropy is equal to the unit entropy:(92)$$\langle S\rangle =S_{i}.$$ With such equality, there would be basically no reason to distinguish entropy from its average value. This situation seems to apply to the Boltzmann-type entropy. This raises the question about the relationship between the Gibbs entropy and the Boltzmann entropy, which will be investigated further.

If, in general Definition 10 of the Boltzmann $S_{B}$ entropy, we assume that each of the W states has the same probability, then it must be that $p_{i}=1/W$. Then, the possible averaging of the Boltzmann entropy is trivial:(93)$$\langle S_{B}\rangle =\sum_{\left\{W\right\}}p_{i}S_{B}=\sum_{\left\{W\right\}}\frac{1}{W}k_{B}\ln W=W\cdot \frac{1}{W}k_{B}\ln W=k_{B}\ln W=S_{B}.$$ Everything has been extensively described here for full transparency.

A similar result should also apply to entropy of the $S_{BT}$-type (Definition 11). However, it is not that obvious. The calculation of the $S_{BT}$ entropy was based on the Maxwell–Boltzmann distribution, which is not a distribution of equal probabilities. However, this is a completely different sense of the division of states from the canonical distribution for the $S_{F}$ entropy (see below). In the case of $S_{BT}$, the states of scattered *N* particles were counted and were homogeneous in three-dimensional physical space and isotropic in three-dimensional phase space. All these microstates corresponded to the same-looking macrostate and had the same probabilities. However, the Maxwell–Boltzmann distribution influenced the non-uniform but isotropic distribution of particles in three-dimensional phase space. Therefore, this distribution influenced the form of states but did not differentiate the probabilities of these very similar states.

The same property of equal probabilities, i.e., the equality of mean entropy with entropy, should also apply to the entropy of the $S_{BE}$-type (Definition 12). No specific calculation of this entropy (or result) was provided, only an outline of how to understand the calculation of such entropy. Contrary to appearances, this is not the same calculation as the entropy of the $S_{\Gamma }$-type calculated in the microcanonical system. In the $S_{BE}$ entropy, similarly to the $S_{BT}$ entropy, homogeneous states in the cells of three-dimensional position space are considered (and by default, they are supposed to be isotropic in three-dimensional phase space). Calculations in position space should proceed as in the $S_{BT}$ entropy, but calculations in three-dimensional phase space are problematic. One should use the energy partition function in a vague way while maintaining the isotropicity and non-uniformity of the velocity distribution. Perhaps, Definition 12 is not fully defined. Nevertheless, it is justified to analyze it in order to show the problem and ideological differences in the ways of understanding entropy. The simplest way would be to perform energy partition calculations in configuration phase space (without position coordinates) in a manner analogous to $S_{\Gamma }$. The $S_{BE}$ entropy understood in this way would be a hybrid of the $S_{BT}$ entropy and the $S_{\Gamma }$ entropy.

Therefore, it is worth taking a closer look at the $S_{\Gamma }$ entropy in terms of elementary probabilities and in terms of entropy averaging. The Gibbs divider N! appears here, which somewhat arbitrarily enters as a divisor of the phase volume of the configuration space $W_{\Gamma }/N!$ expressed in elementary units ω. Consequently, this factor also enters the configuration-phase volume differential $dW_{\Gamma }^{\prime }/N!$. The microcanonical distribution is a uniform continuous distribution (with probability density $\rho =N!/W_{\Gamma }$), so the average Boltzmann entropy of the $S_{\Gamma }$-type is expressed as follows:(94)$$\langle S_{\Gamma }\rangle =\int_{0}^{W_{\Gamma }}\rho \;S_{\Gamma }^{\prime }\frac{dW_{\Gamma }^{\prime }}{N!}=\int_{0}^{W_{\Gamma }}\frac{N!}{W_{\Gamma }}k_{B}\ln W_{\Gamma }^{\prime }\frac{dW_{\Gamma }^{\prime }}{N!}=\frac{k_{B}}{W_{\Gamma }}{\left[W_{\Gamma }^{\prime }\ln W_{\Gamma }^{\prime }-W_{\Gamma }^{\prime }\right]}_{0}^{W_{\Gamma }},$$
(95)$$\langle S_{\Gamma }\rangle =k_{B}\ln W_{\Gamma }-k_{B}=k_{B}\ln\frac{W_{\Gamma }}{e}=S_{\Gamma }-k_{B}\approx S_{\Gamma }.$$

Therefore, practically, the average entropy of the $S_{\Gamma }$-type is equal to this entropy. In addition, it is easy to see that this result does not need to use the N! Gibbs divider.

If entropy has a property 〈S〉=S, this may mean that it is the expected value of some more elementary quantity, e.g., S=〈Y〉. Then, it is obvious that 〈S〉=〈〈Y〉〉=〈Y〉=S. This situation occurs, for example, for the Gibbs $S_{G}$ entropy. This also applies to the $S_{F}$ entropy based directly on the canonical system.

**Statement** **5** (Equivalence of Gibbs and $S_{F}$ Entropies)**.***The* $S_{F}$ *entropy given in the canonical system by Definition 15 can be written as the Gibbs entropy given by Definition 14:*(96)$$S_{F}=-{\left(\frac{\partial F}{\partial T}\right)}_{V,N}=-k_{B}\sum_{i}p_{i}\ln p_{i}=S_{G}.$$

**Proof.** In the canonical probability distribution, the data are given by the expression
(97)$$p_{i}=\frac{1}{Z}\exp\left(-\frac{E_{i}}{k_{B}T}\right),$$
whose logarithm is
(98)$$\ln p_{i}=-\ln Z-\frac{E_{i}}{k_{B}T}.$$ Therefore, the Gibbs entropy is expressed as follows:
(99)$$S_{G}=k_{B}\sum_{i}p_{i}\ln Z+\sum_{i}p_{i}\frac{E_{i}}{T}=k_{B}\ln Z+\frac{\langle E\rangle }{T}.$$ Let us transform the second part of the expression:
(100)$$\sum_{i}p_{i}\frac{E_{i}}{T}=\sum_{i}\frac{1}{Z}\frac{E_{i}}{T}\exp\left(-\frac{E_{i}}{k_{B}T}\right)=\frac{k_{B}T}{Z}\frac{\partial }{\partial T}\sum_{i}\exp\left(-\frac{E_{i}}{k_{B}T}\right),$$
which, thanks to the definition of a statistical sum, simplifies into
(101)$$\sum_{i}p_{i}\frac{E_{i}}{T}=\frac{k_{B}T}{Z}\frac{\partial }{\partial T}Z=k_{B}T\frac{\partial }{\partial T}\ln Z.$$Therefore, the Gibbs entropy can be written as
(102)$$S_{G}=k_{B}\ln Z+k_{B}T\frac{\partial }{\partial T}\ln Z=\frac{\partial }{\partial T}\left(k_{B}T\ln Z\right),$$
which is already the expression for the entropy of the $S_{F}$-type:
(103)$$S_{G}=-{\left(\frac{\partial F}{\partial T}\right)}_{V,N}=S_{F}.$$□

The thesis of the above Statement 5 is not original news. However, this thesis is not completely obvious, so for the sake of clarity, it had to be provided. Moreover, the entropy of the $S_{F}$-type suggests that the Helmholtz free energy *F* or the quantity −F/T may play a role similar to entropy. In any case, the latter quantity is expressed in terms of the statistical sum *Z*, the partition function, just as the general Boltzmann entropy is expressed in terms of W.

### 3.2. Comparison of Statistical Definitions of Entropy and Study of Its Additivity on Simple Examples

There is no reason to consider the Boltzmann and Gibbs (or Gibbs–Shannon) entropies a priori as identical. However, the standard approach considers them to be equivalent in some sense. There is also a common view that the Gibbs formula is more general. The analysis of specific examples shows that Boltzmann entropies sometimes seem to vary by a factor of *N* or by a divisor of *N* within the logarithm. This second aspect is related to the problem of entropy additivity, the Gibbs paradox or, more simply, the Gibbs divisor N!.

We will try to look at these contexts (entropy comparison, entropy additivity and Gibbs divisor) in a few possibly simplified examples. For clarity, let us list these examples in advance, giving the context they are intended to serve:Expample 1.The case of an equal probability distribution. In such a distribution, the Boltzmann and Gibbs definitions give a consistent entropy value.Expample 2.The analysis of entropy and the probability of finding a gas particle distributed in a specific way in the cells of space. The Boltzmann entropy here is *N* times the Gibbs entropy.Expample 3.The Boltzmann entropy of the uniform distribution of gas particles in the cells of space. The entropy additivity problem due to the lack of a divisor *N* under the logarithm sign.Expample 4.The application of the Gibbs divisor N! to explain the additivity problem for spatial cells and its standard interpretation. The Gibbs divisor N! introduces the divisor *N* under the sign of the logarithm of entropy.Expample 5.An unsuccessful alternative attempt to introduce a Gibbs divisor for a hybrid non-equilibrium arrangement of particles in space cells. Trying to explain the problem of additivity as a single argument seems to make sense, but it fails in a detailed calculation.Expample 6.A presentation of the additivity problem using the example of permutations. A simple example examining the basic importance of a problem like the Gibbs paradox.Expample 7.The presentation of the entropy additivity problem with the example of a table with four legs. The suggestion that perhaps entropy counts different material configurations, rather than merely counting the possibilities leading to the same configuration. This would be a reinterpretation of the quantum approach of indistinguishable particles.

For Example 1, consider a uniform probability distribution based on equal probabilities. The equality of the probabilities of elementary events is called classical probability (not in the context of classical vs. quantum). Therefore, if a given macrostate has a realization in W possible equally probable microstates, then the probability of the microstate is $p_{i}=1/W$. Let us apply the Gibbs formula to this situation:(104)$$S_{G}=-k_{B}\sum_{\left\{W\right\}}p_{i}\ln\;p_{i}=-k_{B}W\cdot \frac{1}{W}\ln\frac{1}{W}=k_{B}\ln W=S_{B}.$$ An example of this type is often given (see [34]) and has already appeared almost once in this work in formula (Equation 93). At that time, however, it was about the expected value of the Boltzmann entropy, and not the relationship with the Gibbs entropy. However, it so happened that in Example 1, both contexts overlap. This, however, does not end the analysis of the compatibility of the Boltzmann and Gibbs entropies.

As Example 2, consider the arrangement of *N* (“numbered”) particles in *n* cells with $k_{i}$ particles in each cell, which corresponds to the condition
(105)$$k_{1}+k_{2}+k_{3}+\cdots +k_{n}=N.$$ The number of all possibilities for such arrangement corresponds to the product of the combinations of filling subsequent cells:(106)W=Nk1N−k1k2N−k1−k2k3⋯knkn. This, when written out, corresponds to the number of permutations with repetitions
(107)$$W=\frac{N!}{k_{1}!k_{2}!k_{3}!...k_{n}!}.$$

By using Stirling’s formula for large numbers ($\ln k_{i}!\approx k_{i}\ln k_{i}-k_{i}$), we can transform the logarithm of the number of all states into the form
(108)$$\ln W\approx N\ln N-N-\sum_{i}^{n}\left(k_{i}\ln k_{i}-k_{i}\right)=-\sum_{i}^{n}k_{i}\ln\frac{k_{i}}{N}.$$ If we now interpret $k_{i}/N$ as the probability $p_{i}$ of finding a randomly selected particle in the *i*th cell (equal to 1/n for equal cells), i.e.,
(109)$$p_{i}:=\frac{k_{i}}{N},\;\;\;\;\sum_{i}^{n}p_{i}=1,$$
then the Boltzmann entropy will become similar to the Gibbs entropy:(110)$$S_{B}=k_{B}\ln W\approx -k_{B}N\sum_{i}^{n}p_{i}\ln p_{i}=NS_{G}>S_{G}.$$

However, what is missing for consistency here is the directly multiplicative number *N* in the Gibbs formula. The result of Example 2 should be considered only as an example (perhaps imperfect) showing that the Boltzmann and Gibbs entropies are not the same but not necessarily different (see previous Example 1 and Formula (Equation 104)). In any case, already in the introductory section (chapter), it was signaled that the Boltzmann entropy of the $S_{H}$-type (Definition 16) is basically conceptually defined as *N* times greater than the Gibbs $S_{G}$ entropy (Definition 14).

Example 3 is basically a special case of Example 2, but it is related to the volume fraction of the entropy of an ideal gas. Let us, therefore, consider the Boltzmann entropy as in Example 2 but assume that *n* cells including *N* particles are uniformly distributed:(111)$$S_{B}\approx k_{B}N\ln n=k_{B}N\ln\frac{V}{\upsilon },$$
where υ is the volume of one cell, with all cells together having a volume of *V*. For the full correctness of the volumetric entropy of an ideal gas, the divisor *N* under the logarithm is missing. With this factor, the expression under the logarithm would be intensive, and the entropy would be extensive—it would meet the condition of additivity. In Example 3, the Gibbs entropy, which is completely devoid of the *N* variable, is also non-additive:(112)$$S_{G}\approx k_{B}\ln n=k_{B}\ln\frac{V}{\upsilon }.$$

As Example 4, the same physical system as in Example 2 and Example 3 is considered, but the full set of possible arrangement of *N* molecules in *n* cells is considered. In addition, it is demonstrated how the Gibbs divisor N! works here. Taking into account all possibilities of variation with repetitions would lead to a Boltzmann-type entropy such as in Example 3:(113)$$\ln W=\ln n^{N}=N\ln n=N\ln\frac{V}{\upsilon }.$$

However, this result is not approximate, and it is calculated for all states, including non-equilibrium (non-homogeneous) states. Again, this form of entropy still does not contain a divisor *N* under the logarithm. Therefore, a Gibbs divisor N! is introduced for the number of states, which is interpreted in terms of the indistinguishability of quantum particles:(114)$$\ln\frac{n^{N}}{N!}=N\ln n-\ln N!\approx N\ln n-N\ln N+N=N\ln\frac{n}{N}+N.$$ If we ignore the negativity of the first (main) term of the above result, the Boltzmann entropy can be written as
(115)$$S_{B}\approx Nk_{B}\ln\frac{V}{N\upsilon }+Nk_{B}.$$

This result is consistent with the volume term of the ideal-gas entropy, even including an additive term proportional to *N*. A factor of 1 in this term increased by a factor of 3/2 from the kinetic part of the entropy leads to a total factor of 5/2.

In Example 5, for the same physical system as in Example 4 (as well as 2 and 3), we try to provide an alternative to the Gibbs divisor N!. The procedure is, in a sense, hybrid. First, we assume possible arrangements as at the beginning of Example 4. Then, we reduce the number of possibilities, not as in Example 4, but as in Example 2, by using the equal distribution of particles of Example 3. We consider more non-equilibrium cases than in Example 3, and the most numerous equilibrium cases are reduced, as in Example 2 or 3, by a divisor whose logarithm can be expressed as follows:(116)$$\ln{\left(k!\right)}^{n}\approx nk\ln k-nk=N\ln\frac{N}{n}-N=N\ln N-N\left(1+\ln n\right).$$

If the last term could be omitted, the logarithm of this divisor would be equal to the logarithm of the Gibbs divider:(117)$$\ln{\left(k!\right)}^{n}∼N\ln N∼\ln N!.$$

Unfortunately, a full (though hybrid) account of the number of states with the considered reduction in states does not give the desired result:(118)$$\ln\frac{n^{N}}{{\left(k!\right)}^{n}}=N\ln n-n\ln k!\approx N\ln n-nk\ln k+nk=N\ln\frac{n^{2}}{N}+N.$$

Here, the proof is spoiled by the square of the number of cells, which translates into the square of the volume. So, Example 5 gives a negative result, but at the same time, it is an interesting attempt to solve the Gibbs divisor problem.

In Example 6, we consider 2N particles occupying 2N individual positions, with this arrangement being clearly divided into two halves. The Boltzmann entropy of both halves and the Boltzmann entropy of the whole result from the number of permutations of the particle positions (let us assume $k_{B}:=1$) are
(119)$$S_{1}=S_{2}=\ln N!\approx N\ln N-N,\;\;\;\;S=\ln\left(2N\right)!\approx 2N\ln\left(2N\right)-2N.$$

We, therefore, see that the classical Boltzmann entropy of the whole is greater than the sum of the entropies of the halves, i.e.,
(120)$$S\approx S_{1}+S_{2}+2N\ln2∼S_{1}+S_{2},$$
with the additive term which perhaps could be neglected for a large value of *N*. Therefore, on one hand, Example 6 highlights the entropy additivity problem, and on the other hand, it shows that it may not be that deep quantitatively for large numbers of particles.

Finally, let us visually illustrate the additivity problem of Example 7 based on the example of a macroscopic table. The table has four congruent legs that can be mounted at any corner. The question is as follows: do the theoretical 4!=24 leg assembly possibilities affect the table’s entropy? Or alternatively, does one final shape of the table determine zero entropy? The first approach does not provide entropy additivity for the table halves (left and right), and the second solution is based on material configurations, not the way they are implemented. Formally, half of the table would have an entropy of ln2, so the entire table should have an entropy of 2ln2=ln4≈1.4, while in Boltzmann calculus, the entire table has an entropy of ln24≈3.2, so over twice as much. Perhaps, defining statistical entropy for a system of only four floating elements does not make sense, but the proposed Example 7 illustrates the essence of a material configuration analogous to the quantum energy configuration or to the assumption of the quantum indistinguishability of particles.

### 3.3. The Statistical Nature of Entropy Growth and Its Reservation in a Simple Probabilistic Scheme

After an extensive review and careful study of the Second Law of Thermodynamics in the language of thermodynamics and in the language of statistical mechanics, we are ready to give a probabilistic formulation of this principle that is, in some sense, free from the paradox of reversibility and yet contains the concept of entropy increase. Strangely enough, like the Inequality of the Second Law of Thermodynamics (Theorem 3) or the more recent Newer Fluctuation Theorem (Theorem 2), the formulation takes the form of something like a mathematical theorem that is subject to proof, even though it is essentially based on the definitions introduced. Perhaps, the simple mathematical framework of the formulation limits its direct references somewhat, but it seems that the formulation reflects the most important elements of the Second Law of Thermodynamics and solves its main paradoxes (e.g., Loschmidt’s paradox).

We start with the statistical definition of the elementary Boltzmann entropy, which has appeared before, but in a slightly different context, so it is worth specifying it in this subsection.

**Definition 22** (Elementary Entropy of a Macrostate)**.**
*If the distribution of probabilities of macrostates of a system is defined in terms of the numbers of microstates ($p_{i}=w_{i}/W$), then elementary entropy is understood as a quantity proportional to the logarithm of the number of microstates $w_{i}$ corresponding to a given macrostate, i.e.,*

(121)
$$S:=k_{B}\ln w_{i}=k_{B}\ln\left(p_{i}W\right)=:S_{i},$$

*and as the notations suggest, we limit ourselves to discrete distributions in order to clarify the states and simplify the issue. The number of all microstates corresponding to all macrostates is denoted here by W.*


Interestingly, the above entropy works better than the elementary Gibbs entropy ($-k_{B}\ln p$), because it has the desired monotonicity (it increases with the increase in the probability of state). Meanwhile, the function (−lnp) is decreasing, and the function (−plnp) increases to a maximum and then decreases. Thus, in such a representation, the Gibbs entropy appears to have an artificially changed sign (apart from the additive constant) with respect to the Boltzmann entropy and would not be appropriate for a probabilistic scheme. In other words, using the elementary Gibbs entropy in the probabilistic scheme below would generate a tendency for entropy to decrease rather than increase.

To formulate a clear probability scheme for entropy, it is necessary to use such elementary entropy to be able to use the concept of its average. We associate the average entropy with the formally corresponding values of probability and the number of microstates. We conventionally call these values equilibrium values.

**Definition 23** (Equilibrium Number of States and Equilibrium Probability)**.**
*The values of the number of microstates $\bar{w}$ and the probability $\bar{p}$ corresponding (from the entropy formula) to the value of the average entropy, where*

(122)
$$\langle S\rangle =\sum_{i}p_{i}S_{i}=:k_{B}\ln\bar{w}=:k_{B}\ln\left(\bar{p}W\right),$$

*are called equilibrium values. Due to the discrete probability distribution, the values do not have to be exactly equal to the values corresponding to a particular state (or states). Let us assume that these values correspond to the closest available values of $w_{i}=:{\bar{w}}_{+}$ and $p_{i}=:{\bar{p}}_{+}$, which are not less than $\bar{w}$ and $\bar{p}$, respectively.*


The equilibrium values introduced in this way, based on the mean entropy, should not be confused with the directly averaged values:$$\langle w\rangle :=\sum_{i}p_{i}w_{i}=\frac{1}{W}\sum_{i}w_{i}^{2}\leq w_{j}=:\langle w_{+}\rangle \;\;\;\;,\;\;\;\;\langle p\rangle :=\sum_{i}p_{i}^{2}\leq p_{j}=:\langle p_{+}\rangle ,$$
where the plus signs additionally mark values from the domain of the distribution of the random variable, which are the closest possible values not smaller than the mean value.

Both types of average values and corresponding states in this probabilistic scheme are substitutes for the equilibrium state. At this stage, however, it has more in common with static equilibrium in terms of centers of mass for the probability distribution (on the argument axis and on the value axis), rather than with thermodynamic equilibrium. Theoretically, a more suitable candidate for the thermodynamic equilibrium state is the state of maximum probability and maximum entropy (the elementary Boltzmann entropy is an increasing function). Nevertheless, the constructed scheme has interesting properties.

**Proposition 3** (Probabilistic Scheme of the Second Law of Thermodynamics)**.**
*We assume a discrete non-trivial probability distribution, i.e., one in which at least two probabilities are different. The expected value of the change in the elementary entropy in a state below equilibrium (which has a probability less than the equilibrium probability) is positive:*

(123)
$$\langle S-S(p_{0})\rangle >0\;\;\;\;for\;\;\;\;p_{0}<\bar{p}.$$


*However, in a state with probability no less than the equilibrium probability, the expected value of the entropy change is non-positive:*

(124)
$$\langle S-S(p_{0})\rangle \leq 0\;\;\;\;for\;\;\;\;p_{0}\geq \bar{p}.$$

*In particular, the expected value of the entropy change in the average state (relative to the state with average probability) is even negative, i.e.,*

(125)
$$\langle S-S(p_{0})\rangle <0\;\;\;\;for\;\;\;\;p_{0}=\langle p\rangle \;\;or\;\;p_{0}=\langle p_{+}\rangle ,$$

*which additionally means that the average probability is sharply higher than the equilibrium probability $\langle p_{+}\rangle \geq \langle p\rangle >\bar{p}$ (although $\langle p_{+}\rangle$ and ${\bar{p}}_{+}$ may be equal). However, it turns out that the total expected value of the entropy change taking into account the simple average as well as averaging over the initial reference state is equal to zero:*

(126)
$${\langle \langle S-S_{0}\rangle \rangle }_{0}=0.$$



As already mentioned, the above formulation of the Probabilistic Scheme of the Second Law of Thermodynamics does not have only the character of a definition or principle that can only be equivalent to another or result from another (or vice versa) principle but is a mathematical theorem (although simple and firmly rooted in the preceding definitions). Therefore, one can provide a proof of it without making an additional statement about the relation of equivalence or implication with another principle.

**Proof.** (Probabilistic Scheme of the Second Law of Thermodynamics). We start with the simplest last equality:
(127)$${\langle \langle S-S_{0}\rangle \rangle }_{0}=\sum_{i}\sum_{j}p_{i}p_{j}\left(\ln p_{i}-\ln p_{j}\right)=0,$$
where the elements of the sums subtracting from zero differ only in the conversion of indicators *i* to *j*. The elementary entropy is a concave function (logarithm) of $w_{i}$ or $p_{i}$, so Jensen’s inequality applies to it for non-trivial coefficients (being probabilities $p_{i}\neq 0$ and $p_{i}\neq 1$), and it is a sharp inequality:
(128)$$\sum_{i}p_{i}S\left(p_{i}\right)<S\left(\sum_{i}p_{i}p_{i}\right).$$ The left side is the average entropy (expected value), and the argument of the right side is the average probability:
(129)$$\langle S\rangle <S\left(\langle p\rangle \right)\leq S\left(\langle p_{+}\rangle \right),$$
where Jensen’s inequality is sharp here due to the fact that the average probability of the non-trivial distribution takes a value that is neither the smallest nor the largest probability value. Moreover, the inequality corresponding to the increasing entropy function was also used for $\langle p\rangle \leq \langle p_{+}\rangle$. After a transformation, we obtain the penultimate inequality of Proposition 3:
(130)$$\langle S\rangle -S\left(\langle p\rangle \right)=\langle S-S\left(\langle p\rangle \right)\rangle <0.$$The inequalities for probabilities smaller or larger than equilibrium are more trivial and result from the monotonicity of the increasing entropy function with respect to the corresponding values for the mean entropy, i.e.,
(131)$$\langle S\rangle >S\left(p_{0}<\bar{p}\right),$$
(132)$$\langle S\rangle \leq S\left(p_{0}\geq \bar{p}\right),$$which proves the first and the second inequalities and thus ends the proof of Proposition 3. □

The main conclusions of Proposition 3 are that entropy tends to increase but in the sense of an average value and with respect to an appropriate initial state. Moreover, it shows that the law of entropy increase is not absolute and may not apply, firstly, in a state of high probability (at least equilibrium)—even in the sense of an average value—and secondly, in transitions to low probability states—apart from averaging, which obscures the existence of such transitions. Indeed, from other formulations, we know that the law of increasing entropy does not apply to equilibrium states, which can fluctuate to lower entropy.

Perhaps, more clear than the proof and theoretical analysis of the Probabilistic Scheme of the Second Law of Thermodynamics will be the example of a simple system analyzed in the last subsection of this article before the conclusion. Examples with larger numbers of particles and states would reveal a general tendency for average and equilibrium probabilities to be close to maximum probabilities. Unfortunately, it has not yet been possible to formulate this noticeable tendency in terms of a theorem within the Probabilistic Scheme of the Second Law of Thermodynamics.

### 3.4. Demonstration of Probabilistic Scheme of Second Law of Thermodynamics with a Simple Example

The methods of statistical physics generally make sense for large numbers of particles—say of the order of the Avogadro constant, i.e., $10^{23}$. However, we are not able to study such large number of particles in detail, so it is worth doing it on a small number of particles. For example, Reif’s textbook gives many diagrams drawn by using computer simulations for 40 (or even just 4) balls that can be located in only two neighboring cells of space [70].

Let us consider a very simple system consisting of three balls (N=3) that can be arranged in three cells (n=3). A given cell can be empty (k=0), contain one ball (k=1), two (k=2) or three balls (k=3). Therefore, the number of all microstates is $W=n^{N}=3^{3}=27$. However, the macrostate is determined only by the number (cardinality) of balls in the cells, not by the set of individual numbers of these balls.

Before we learn the simplified designations of these macrostates, we will take a look at one of them. Let there be a state in which there is one ball in the first cell, the second cell is empty, and the third cell has two balls: 0 00. In the graphic diagram, each ball is marked with the number zero. The considered macrostate is represented by three microstates (w=3), which can be presented in graphical diagrams containing ball numbers: 1 23, 2 13 and 3 12. Of course, the order of the two balls in a given cell does not matter.

The easiest way to mark macrostates is with three digits, which are the numbers of balls in individual cells. For example, the macrostate considered above will be marked “102”. Let us note this along with all ten macrostates:

$\begin{array}{lll} 000 & & \end{array}\;\equiv \;300$, $\begin{array}{lll} & 000 & \end{array}\;\equiv \;030$, $\begin{array}{lll} & & 000 \end{array}\;\equiv \;003$, $\begin{array}{lll} 00 & 0 & \end{array}\;\equiv \;210$,

$\begin{array}{lll} 0 & 00 & \end{array}\;\equiv \;120$, 00 0 ≡ 201, 0 00 ≡ 102, $\begin{array}{lll} & 00 & 0 \end{array}\;\equiv \;021$, $\begin{array}{lll} & 0 & 00 \end{array}\;\equiv \;012$, $\begin{array}{lll} 0 & 0 & 0 \end{array}\;\equiv \;111$.

The main parameter of each macrostate is the corresponding number of microstates *w*. If the number of all microstates of the system is W=27, then the probability of a given state is
(133)$$p=\frac{w}{W}=\frac{w}{27}.$$

However, the Boltzmann-type entropy for macrostates can be calculated by omitting the Boltzmann constant ($k_{B}:=1$):(134)S=lnw.

The parameters of all macrostates are presented in Table 1. Based on this table, a matrix of elementary entropy differences is shown in Table 2. However, for calculating average values, the partial entropy difference matrix taken with the state probability weight shown in Table 3 is more useful. Therefore, on this basis, the most important table, Table 4, can be made.

It is also worth providing the average number of microstates and the average probability, i.e.,
〈w〉=3.444,〈p〉=0.128,
and similarly the equilibrium values related to the mean entropy,
$$\langle S\rangle =1.131,\;\;\;\bar{w}=3.097,\;\;\;\bar{p}=0.115.$$

From Table 1, we conclude that the above average and equilibrium values are only lower than the value corresponding to the state 111 with the highest probability. In practice, this means that in the example considered, the state assigned to the equilibrium state is the state with the highest probability:$$0.111<{\bar{p}}_{+}=\langle p_{+}\rangle =p_{max}=0.222.$$

As the last table (Table 4) shows, in the example considered, for as many as 9 out of 10 states, the expected value of the entropy change is positive. Only for the most probable state, the entropy increase is negative (i.e., entropy decreases). This reflects well the statistical nature of the Second Law of Thermodynamics but also shows that it is not an absolute principle (one average change in entropy is negative). Thanks to this negative change, the total average of all entropy changes is zero, which also helps explain the paradox of irreversibility.

Note that the Probabilistic Scheme of the Second Law of Thermodynamics does not use the concept of time. The scheme only describes the probabilities of random events, as in a board game with dice rolls. In a sense, we can imagine discrete time here as a sequence of successive random samples consistent with a probability distribution. Although the probability distribution in the example (Table 1) is stationary, there is a certain preference for generating a positive entropy increase during transitions between probability states (Table 4), which we can interpret as a “temporal” evolution towards more probable states. Of course, the Probabilistic Scheme of the Second Law of Thermodynamics needs to be tested further for examples with larger numbers of particles, but for now, it is only given as Proposition 3.

## 4. Conclusions

Based on a broad review of the issue and thorough analysis, three propositions for the Second Law of Thermodynamics of thermodynamics were formulated. Two of them concern phenomenological thermodynamics, and one concerns statistical mechanics. Propositions within thermodynamics are not based on the concept of entropy. Proposition 1 corrects the non-existence of the *Perpetuum Mobile* of the Second Kind in the Kelvin–Ostwald Statement to the more restrictive non-existence of the *Perpetuum Mobile* of the Third Kind. Proposition 2 reinforces the global principle of no heat flow towards any higher temperature (Clausius’ First Statement) with a stronger Heat–Temperature Inequality. Proposition 3, within the framework of statistical mechanics, uses the concept of elementary probabilistic entropy but does not use its time evolution in the form of some extension of the Liouville equation (such as the kinetic Boltzmann equation). Therefore, this proposal, called the Probabilistic Scheme of Second Law of Thermodynamics, is free from Loschmidt’s irreversibility paradox. In practice, this means that Proposition 3 allows for a negative entropy change, but this is unlikely or requires the system to be in a particular state. However, the rule is that entropy increases but in the sense of the average value and, additionally, provided that the system is not in the state with the highest entropy.

An interesting property that qualitatively differentiates propositions in the phenomenological thermodynamic approach from propositions in the probabilistic approach is the status of evidence. The thermodynamic statements (Propositions 1 and 2) cannot be proven directly, but their equivalence to the principle of non-decreasing entropy can be proven. However, a probabilistic statement (e.g., Proposition 3), as it turns out, is subject to direct proof—but the discussed property does not entitle us to prefer statistical mechanics over phenomenological thermodynamics (or vice versa). The point is that thermodynamic statements are more physical and categorical, while statistical statements are more mathematical and do not contain an absolute and unconditional version of the Second Law of Thermodynamics (because they are not related to fluctuation and to the paradox of irreversibility). Therefore, thermodynamics and statistical mechanics must remain complementary and correspond to each other.

Moreover, this work defines new concepts of non-existent *Perpetuum Mobile* machines—not only the *Perpetuum Mobile* of the Third Kind. At the same time, on the 200th anniversary of Carnot’s work from 1824, it was pointed out that the Carnot engine is not any *Perpetuum Mobile*, so it should not be just an imaginary theoretical model but should be technically feasible in the sense of a machine working on the Carnot comparative cycle. Moreover, in the context of comparing the Heat–Temperature Inequality with the structure of the construction of real numbers given by Eudoxus of Cnidus, it is the Carnot engine that corresponds to rational numbers (defined by equalities, not inequalities) and is therefore a realizable device. In any case, there is currently no working prototype of the Carnot engine, and the two existing patents seem to be far from its implementation, as they do not contain any relevant significant technical solutions. It is wrong to think that heat engines are relics of the past, because, for example, even in nuclear power plants, electricity is based on steam turbines.

## Figures and Tables

**Table 1 entropy-26-01122-t001:** List of macroscopic states of a system of three balls in three cells, along with their number of microstates *w*, probability *p* and elementary Boltzmann entropy *S*.

State →	300	030	003	210	120	201	102	021	012	111
*w*	1	1	1	3	3	3	3	3	3	6
*p*	0.037	0.037	0.037	0.111	0.111	0.111	0.111	0.111	0.111	0.222
*S*	0	0	0	1.099	1.099	1.099	1.099	1.099	1.099	1.792

**Table 2 entropy-26-01122-t002:** The matrix of relative entropy differences $\Delta S=S_{i}-S_{j}$ of elementary macrostates of the three-ball system in three cells (*i* refers to the states in the first column, and *j* refers to the states in the first row).

${i↓}^{j\to }$	300	030	003	210	120	201	102	021	012	111
300	0	0	0	−1.099	−1.099	−1.099	−1.099	−1.099	−1.099	−1.792
030	0	0	0	−1.099	−1.099	−1.099	−1.099	−1.099	−1.099	−1.792
003	0	0	0	−1.099	−1.099	−1.099	−1.099	−1.099	−1.099	−1.792
210	1.099	1.099	1.099	0	0	0	0	0	0	−0.693
120	1.099	1.099	1.099	0	0	0	0	0	0	−0.693
201	1.099	1.099	1.099	0	0	0	0	0	0	−0.693
102	1.099	1.099	1.099	0	0	0	0	0	0	−0.693
021	1.099	1.099	1.099	0	0	0	0	0	0	−0.693
012	1.099	1.099	1.099	0	0	0	0	0	0	−0.693
111	1.792	1.792	1.792	0.693	0.693	0.693	0.693	0.693	0.693	0

**Table 3 entropy-26-01122-t003:** Probability-weighted matrix of relative elementary entropy differences $p_{i}\left(S_{i}-S_{j}\right)$ of macrostates of the system of three balls in three cells (*i* refers to the states in the first column, and *j* refers to the states in the first row).

${i↓}^{j\to }$	300	030	003	210	120	201	102	021	012	111
300	0	0	0	−0.041	−0.041	−0.041	−0.041	−0.041	−0.041	−0.066
030	0	0	0	−0.041	−0.041	−0.041	−0.041	−0.041	−0.041	−0.066
003	0	0	0	−0.041	−0.041	−0.041	−0.041	−0.041	−0.041	−0.066
210	0.122	0.122	0.122	0	0	0	0	0	0	−0.077
120	0.122	0.122	0.122	0	0	0	0	0	0	−0.077
201	0.122	0.122	0.122	0	0	0	0	0	0	−0.077
102	0.122	0.122	0.122	0	0	0	0	0	0	−0.077
021	0.122	0.122	0.122	0	0	0	0	0	0	−0.077
012	0.122	0.122	0.122	0	0	0	0	0	0	−0.077
111	0.398	0.398	0.398	0.154	0.154	0.154	0.154	0.154	0.154	0

**Table 4 entropy-26-01122-t004:** List of expected values of entropy differences across states, including the total expected value (equal to zero, but in the summation, errors of approximation accumulate—marked with underline).

State →	300	030	003	210	120	201	102	021	012	111
〈ΔS〉	1.131	1.131	1.131	0.032	0.032	0.032	0.032	0.032	0.032	−0.661
Sum					2.924					
p〈ΔS〉	0.042	0.042	0.042	0.004	0.004	0.004	0.004	0.004	0.004	−0.147
${\langle \langle \Delta S\rangle \rangle }_{0}$					0.003					

## Data Availability

No new data were created or analyzed in this study. Data sharing is not applicable to this article.

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
