# Peer review of "Revisions of the Phenomenological and Statistical Statements of the Second Law of Thermodynamics"

_entropy, 2024, doi:10.3390/e26121122_

Round 1
Reviewer 1 Report
Comments and Suggestions for Authors
The review paper by Koczan and Zivieri tries to reveal the pitfalls of (non)equilibrium thermodynamics and covers the passage between its phenomenological and statistical descriptions. While the first attempt at it by Koczan, published in Entropy two years ago, was a useful appearance (see ref. 14 of the current ms.), its extension by the present paper could be viewed as a more logical puzzle-solving than the useful ideas-generating formulation of thermodynamic subtleties, exhaustively discussed elsewhere.
Thermodynamics after all is an engineering science, see the seminal book by Moran and Shapiro titled Fundamentals of Engineering Thermodynamics to name but one to be taken into account for the considerations presented in the present study.
In the present reviewer's opinion, the paper provides lengthy and obvious repetitions of phenomenology as well as statistical descriptions of thermodynamics' fundamentals in a form that does not provide a new context/insight. Moreover, the paper's content is not based on solid (and, lucid) physical examples, such as the ones, that questioned the Boltzmann H-theorem, or the other ones, unveiling some hard-to-treat physical systems - the glassy or colloidal (viz soft-matter) ones, to mention but two; this would be a real challenge of the considerations presented. In the end, the introduction of the review lasts for 14 pages (out of 36; thus about 40% of the total) which is both strange and unacceptable. To conclude, the paper is definitely unsuitable for publication in its present form.
Author Response
Please see the attachment PDF file.

Reviewer 2 Report
Comments and Suggestions for Authors
see attached file

English must be improved
Author Response
Please see the attachment PDF file.

Reviewer 3 Report
Comments and Suggestions for Authors The authors have reviewed the thermodynamic formulations of the second law of thermodynamics and entropy from the perspective of phenomenological thermodynamics and statistical mechanics. It is proven that the two new propositions (1 and 2) are equivalent to the full entropic formulations of the second law of thermodynamics. The third (3) new proposition, within the framework of statistical mechanics, uses the concept of elementary probabilistic entropy without using its time evolution according to the Boltzmann equation. Discussions of various definitions of entropy expressions (phenomenological and statistical) included in a single article are a novel idea towards the understanding of the theory of the second law of thermodynamics. However, I could not find more novelty in this work compared to what has already been published in the past. The authors also fail to reconcile the findings of their theoretical results with the existing theories of entropy. I have the following suggestions to improve the manuscript. #English usage is poor. The following errors came to my notice. Lines 34-35, 53-54, 57-59, 99, 112-113, 118, 136, 203-204, 221, 355-356, 454, 474, 479, 520, 524-525, 551-554, 575, 577, 580, 604, 612, 640, 677, 723, 735, 766, 769, 778, 780-785 (repetition of 770-773), 799, 802, 873, 899, 914, 968, 969, 977, 999, 1080, 1305. # Finish 'The Introduction' at line 124 and change the section numbers accordingly. # All the definitions 1-20 should be presented briefly with one example and limitation in each case because most of these are published results. #Line 192: 'heat' should be replaced by 'reversible heat'. #Line 409: 'energy' should be replaced by 'energy/T'. # Write Eq.(31) correctly. # Line 545: replace 't' with 'T'. # Definition 17 is written twice (starting at lines 541 and 602, respectively). #Line 799: Remove 'and reduce'. #Line 873: Remove 'her' by 'this'. #Lines 882-887: 'This uneven proportion .......irrational number' should be deleted. #Line 899: Remove 'assuming'. #Line 938: Delete 'At the end of this chapter'. #Line 942: Delete' well'. #Line 951: Delete 'well'. #Line 968: Replace 'de fact' with 'in fact'. #Line 969: Delete 'okay' and 'well'. #Line 977: Delete 'of the construction'. #Delete Lines 986-994. #Line 999: Change 'is where' to 'is from where'. #Line 1057: What is S (SE)? #Eq.(106): k2 should be replaced by k3 (right side, third term-bottom). #Eq.(116): Error in calculation: '(nk ln k+nk)' should be '(nk ln k-nk)'. #Line 1305: Delete 'as in the dispute over .....eve'. # Give the following proofs in an Appendix (if required) [Starting at Line numbers 642, 673, 683, 715, 788, 896, 1073, 1236].
Round 2
Reviewer 1 Report
Comments and Suggestions for Authors
Still, a little bit more elaborated examples, such as the one by Moran and Shapiro, ref. 14, are missing a bit in "Revisions of the Phenomenological and Statistical Statements of the Second Law of Thermodynamics" but anyway, the paper is much improved, and the Authors have taken the referee's suggestions quite seriously into account. Thus, the paper can be recommended for publication, especially in terms of nonequilibrium and non-Gaussian statistics. They have properly been included too, ref. 49.
Author Response
We thank Reviewer 1 for the current green light to the manuscript and for earlier comments. The bibliography of the manuscript has been further enriched with the extensive handbook by Rumer and Ryvkin in the context of the BBKGY hierarchy and also the Navier-Stokes equation.
Reviewer 2 Report
Comments and Suggestions for Authors
I don't have further comments.
Author Response
We thank the Reviewer 2 for the current green light for the manuscript and for his earlier comments. We also encourage you to follow the manuscript updates.